# MIB: A Mechanistic Interpretability Benchmark

Aaron Mueller [*1]  Atticus Geiger [*2]  Sarah Wiegreffe [3]  Dana Arad [4]  Iván Arcuschin [5]  Adam Belfki [6]
Yik Siu Chan [7]  Jaden Fiotto-Kaufman [6]  Tal Haklay [4]  Michael Hanna [8]  Jing Huang [9]  Rohan Gupta [10]
Yaniv Nikankin [4]  Hadas Orgad [4]  Nikhil Prakash [6]  Anja Reusch [4]  Aruna Sankaranarayanan [11]  Shun Shao [12]
Alessandro Stolfo [13]  Martin Tutek [4]  Amir Zur [2]  David Bau [6]  Yonatan Belinkov [4]

## Abstract

How can we know whether new mechanistic interpretability methods achieve real improvements? In pursuit of lasting evaluation standards, we propose MIB, a Mechanistic Interpretability Benchmark, with two tracks spanning four tasks and five models. MIB favors methods that *precisely* and *concisely* recover relevant causal pathways or causal variables in neural language models. The **circuit localization track** compares methods that locate the model components—and connections between them—most important for performing a task (e.g., attribution patching or information flow routes). The **causal variable localization track** compares methods that featurize a hidden vector, e.g., sparse autoencoders (SAEs) or distributed alignment search (DAS), and align those features to a task-relevant causal variable. Using MIB, we find that attribution and mask optimization methods perform best on circuit localization. For causal variable localization, we find that the supervised DAS method performs best, while SAE features are not better than neurons, i.e., non-featurized hidden vectors. These findings illustrate that MIB enables meaningful comparisons, and increases our confidence that there has been real progress in the field.

## 1. Introduction

To understand how and why language models (LMs) behave the way they do, we must understand the underlying causes of their behavior. To this end, mechanistic interpretability

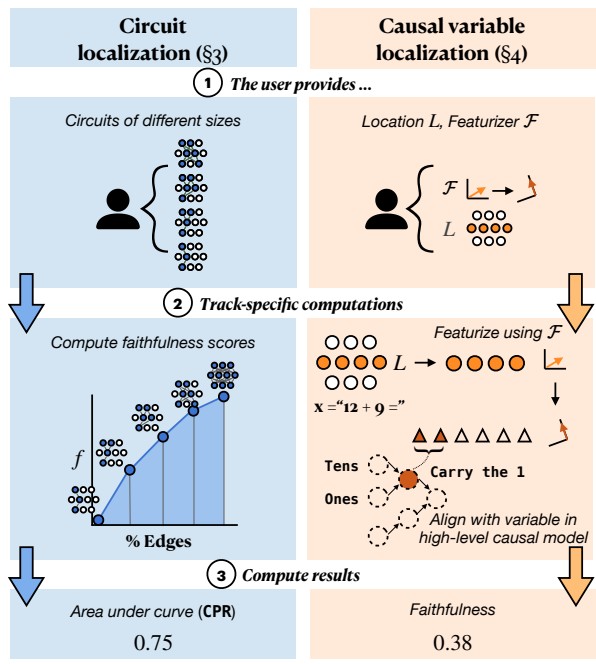

*Figure 1.* Overview of MIB. We compare different circuit (§3) and causal variable (§4) localization methods on their ability to faithfully represent a model's behavior on a given task. We provide standardized datasets and metrics for this purpose, and accept user submissions for display on two public leaderboards.

(MI) methods have proliferated quickly. MI methods can yield deep insights into LM behaviors (Räuker et al., 2023; Ferrando et al., 2024; Sharkey et al., 2025, *i.a.*), and sometimes yield more fine-grained control over LM behaviors than standard training or inference techniques (Meng et al., 2022; Marks et al., 2025). However, it is difficult to directly compare the efficacy of MI methods. New methods are often compared to prior methods via ad hoc evaluations using metrics that may not produce generalizable insights. Thus, how can we know whether new methods are producing real advancements over prior work?

We propose a benchmark to provide a basis for comparisons. A benchmark is a claim as to what should be considered important to a field. In our case, *the ability to precisely locate,*

---

[*]Equal contribution  [1]Boston University  [2]Pr(AI)²R Group  [3]Ai2  [4]Technion  [5]University of Buenos Aires  [6]Northeastern University  [7]Brown University  [8]University of Amsterdam  [9]Stanford University  [10]Independent  [11]MIT  [12]Cambridge University  [13]ETH Zürich. Correspondence to: Aaron Mueller <amueller@bu.edu>.

*Proceedings of the $42^{nd}$ International Conference on Machine Learning*, Vancouver, Canada. PMLR 267, 2025. Copyright 2025 by the author(s).

*and causally validate, task mechanisms or specific concepts in a neural network* is the key goal of (at least some part of) many MI pipelines. In fact, some have argued that causal analysis and localization are what differentiate MI from other types of interpretability work (Mueller et al., 2024; Geiger et al., 2024a; Saphra & Wiegreffe, 2024). Existing benchmarks compare *within* a specific class of methods (Karvonen et al., 2025; Schwettmann et al., 2023), or on specific tasks and models (Arora et al., 2024; Huang et al., 2024a; Miller et al., 2024; Gupta et al., 2024).

We propose MIB to encourage stable standards for comparing *across* MI methods—specifically, localization and featurization methods—in a principled way. MIB encourages evaluation across a standard suite of models, datasets with fixed counterfactual inputs used for interventions (§2), and principled metric definitions—including novel metrics (§3.1). It includes two public leaderboards that accept submissions for evaluation on a private test set (§2.4).

MIB contains two tracks based on two prominent paradigms in mechanistic interpretability: **circuit localization** (§3) and **causal variable localization** (§4). The circuit localization track benchmarks how well methods can locate the most important subset of model components for performing a given task (Cao et al., 2020; Wang et al., 2023; Conmy et al., 2023). The causal variable localization track benchmarks methods for featurizing hidden vectors (e.g., mapping them to an alternative vector space) and selecting features that implement specific causal variables or concepts (Vig et al., 2020; Geiger et al., 2021; 2024a; Mueller et al., 2024).

Beyond standardizing evaluations, MIB yields several scientific insights. For instance, using MIB, we find that attribution and mask optimization methods outperform other approaches to circuit localization. We find that supervised methods provide better features for causal variable localization, but the popular method of sparse autoencoders (SAEs) fails to provide better features than standard dimensions of hidden vectors. This is evidence that (1) there is clear differentiation between methods, and (2) there has been real progress in mechanistic interpretability.

Our datasets are available on HuggingFace. Code is available on GitHub. The leaderboard is hosted at this URL.

## 2. Materials

### 2.1. Tasks

Both tracks evaluate across four tasks. The tasks are selected to represent various reasoning types, difficulty levels, and answer formats. Two of the tasks—Indirect Object Identification and Arithmetic—were chosen because they have been extensively studied, while the others—Multiple-choice Question Answering and the AI2 Reasoning Challenge—

were chosen precisely because they have *not* been studied.[1]

The number of instances in each dataset and split is summarized in Table 5 (App. D). Each task comes with a training split on which users can discover circuits or causal variables, and a validation split on which users can tune their methods or hyperparameters. We also create two test sets per task: public and private. The public test set enables faster iteration on methods. We release the train, validation, and public test sets on Huggingface. The private test set is not visible to users; they must upload either their circuits or their locations and featurizers to an API (see §2.4).

**Indirect Object Identification (IOI).** This is one of the most studied tasks in MI, first proposed by Wang et al. (2023). IOI has sentences like *"When Mary and John went to the store, John gave an apple to ᵤ"*, containing a subject (*"John"*) and an indirect object (*"Mary"*), which should be completed with the indirect object. As even small LMs can achieve high accuracy, it has been well studied (Huben et al., 2024; Conmy et al., 2023; Merullo et al., 2024). We generate 40,000 instances. See App. D.1 for details.

**Arithmetic.** Math-related tasks are common in MI (Stolfo et al., 2023; Nanda et al., 2023; Zhang et al., 2024; Nikankin et al., 2025) and interpretability research more broadly (Liu et al., 2023; Huang et al., 2024b). We follow Stolfo et al. in defining the task as performing operations with two operands of up to two digits each. Given a pair of numbers and an operator, the model must predict the result of the operation, e.g., *"What is the sum of 13 and 25?"*. To create the dataset, we enumerate all possible pairs of one-digit and two-digit numbers and generate queries for addition and subtraction, yielding about 75,000 instances. Following Karpas et al. (2022) and Stolfo et al. (2023), we use six natural language templates for each operand pair to ensure we are isolating robust behavior. See App. D.2 for details.

**Multiple-choice question answering (MCQA).** MCQA is a common task format for LM benchmarks, though only a few MI works have studied it (Lieberum et al., 2023a; Wiegreffe et al., 2025; Li & Gao, 2024). We expand an existing synthetic dataset designed to isolate a model's MCQA ability from any task-specific knowledge (Wiegreffe et al., 2025). We generate 260 instances. All questions have four choices and are about the color of an object, such as: *"Question: A box is brown. What color is a box?\nA. gray\nB. black\nC. white\nD. brown\nAnswer: "* with answer *"D"*.

**AI2 Reasoning Challenge (ARC).** Finally, we analyze the ARC dataset (Clark et al., 2018), which contains grade-school-level multiple-choice science questions. This is a

---

[1]Studying the same tasks can lead to hill-climbing on narrow distributions. Insights from novel models and task settings could verify that previous advancements are real.

representative task for evaluating basic scientific knowledge in LMs (Brown et al., 2020; Jiang et al., 2023; Dubey et al., 2024). Our work presents the first mechanistic investigation of LM performance on this dataset. We follow the dataset's original partition into Easy and Challenge subsets (5000 and 2500 instances, respectively) and analyze them separately, due to models' large accuracy differences on them (Table 1). We maintain the original 4-choice multiple-choice prompt formatting (see App. D.4), making this dataset related in format to, but more challenging than, MCQA.

## 2.2. Counterfactual Inputs

For both MIB tracks, counterfactual interventions on model components[2] provide the basis for all evaluations. Here, components are set to the value they would have taken if a *counterfactual input* were provided. *Activation patching* is a popular term for this method.[3] Several studies (Vig et al., 2020; Geiger et al., 2021; Chan et al., 2022) have argued that this type of intervention is a useful analysis tool because components are only ever fixed to values that they would *actually realize* (as opposed to interventions that add noise or fix components to constants).

In the circuit localization track, activation patching is used to push models towards answering in an opposite manner to how they would naturally answer given the input. Success is achieved in this setting when counterfactual interventions to components outside the circuit minimally change the model's predictions. In the causal variable localization track, activation patching is used to precisely manipulate specific concepts. Success is achieved in this setting when a variable in a causal model is a faithful summary of the role a model component plays in input-output behavior—i.e., interventions on the variable have the same effect as interventions on the model component.

The counterfactual defines the task to a large extent (Miller et al., 2024);[4] thus, it is crucial that the mapping from a dataset instance to its counterfactual counterpart is fixed. For each task, we define a set of counterfactual inputs to help localize model behaviors. Some of these maintain the

---

[2]We use "component" as a generic term to refer to any (part of a) hidden representation in a model. When referring to submodules like full MLP blocks, we refer to the output vectors of these blocks.

[3]The term *activation patching* includes not only interventions from counterfactual inputs, but also interventions that zero out activations, inject noise, or steer activations to off-distribution values (Wang et al., 2023; Zhang & Nanda, 2024); we use this term because we use mean ablations and optimal ablations as baselines in §3.2. Terms like *resampling ablations* (Chan et al., 2022) or *interchange interventions* (Geiger et al., 2021) more narrowly refer to interventions from counterfactual inputs.

[4]For example, given IOI, if the counterfactual entails replacing a name with a randomly selected one, then the task is now to choose the correct indirect object over a random name; this is distinct from simply generating the correct indirect object.

*Table 1.* Model performance (0-shot, greedy generation) for all models and tasks on our public test splits. For results using ranked-choice scoring and more details on evaluation, see App. D.5.

| | IOI | MCQA | Arithmetic | | ARC | |
| | | | (+) | (−) | (E) | (C) |
|---|---|---|---|---|---|---|
| Llama-3.1 8B | 0.71 | 0.92 | 0.96 | 0.88 | 0.93 | 0.79 |
| Gemma-2 2B | 0.83 | 1.00 | 0.65 | 0.43 | 0.79 | 0.59 |
| Qwen-2.5 0.5B | 0.99 | 1.00 | 0.37 | 0.29 | 0.73 | 0.58 |
| GPT2-Small | 0.92 | 0.06 | 0.00 | 0.10 | 0.03 | 0.03 |

same correct answers as the original instances; some do not. For example, the counterfactual inputs for ARC and MCQA have different answer symbols or answer orders than the original instance, which change the correct answer. Others change the semantics of the input, which may or may not change the correct answer.

We provide counterfactual inputs for each instance in the train, validation, and test sets, where the mappings from the original inputs to the counterfactual inputs are fixed to ensure consistency in evaluation. These are provided on Huggingface. See App. D, Tables 6, 8, 9, and 10 for examples and more details.

## 2.3. Models

To provide baselines for our paper and to initialize entries in the public leaderboard, we evaluate a set of open-weight models. Given the pace of the field, any set of models or tasks will be incomplete; we select 4 models that cover a range of model sizes, families, capability levels, and prominence in MI: Llama-3.1 8B (Dubey et al., 2024), Gemma-2 2B (Riviere et al., 2024), Qwen-2.5 0.5B (Yang et al., 2024), and GPT-2 Small (117M, Radford et al., 2019).

For a mechanistic analysis to be meaningful for a given model and task, the model generally should be able to perform the task well. Performance for all models and tasks with 0-shot prompts[5] and greedy decoding is in Table 1; performance on counterfactual inputs is in App. D.5. Gemma 2 and Llama are capable across the board, performing well on MCQA and ARC (Easy). GPT-2 Small has been extensively studied in MI; it is much smaller than the other models we investigate and less capable (and thus may rely on qualitatively different mechanisms). Qwen-2.5 performs well relative to its size but has not yet been extensively studied.

## 2.4. Leaderboard

We have constructed two online leaderboards (one for each track) hosted on Huggingface to receive user submissions

---

[5]In-context learning is itself a behavior worth studying mechanistically, but one that is challenging to disentangle from task-specific behavior, leading many MI studies to use 0-shot prompts.

and display results. We intend for the leaderboards to serve as a living public artifact that both incentivizes progress in MI and advises users of MI tools on the state of the art. More details about the leaderboards are in App. D.6.

We have constructed a submission portal. It will aggregate required information and links to one's materials, where they will be used to run evaluations on the private test set. Users will be able to use the public test set for fast prototyping of new methods; the private test set will be the measure by which the state of the art is benchmarked.

## 3. Circuit Localization Track

The circuit localization track evaluates how well a method can discover causal subgraphs $\mathcal{C}$—more commonly known as **circuits** (Olah et al., 2020)—that localize the mechanisms underlying how a full neural network $\mathcal{N}$ performs a task. We define metrics for comparing circuit localization methods (§3.1) and compare common methods (§3.2).

**Defining circuits.** A circuit $\mathcal{C}$ is a subgraph of the computation graph of $\mathcal{N}$. Its nodes are typically submodules or attention heads (e.g., the layer 5 MLP, or attention head 10 at layer 12); edges are abstract objects that reflect information flow between a pair of nodes.

### 3.1. Circuit Metrics

The metrics used to evaluate circuits are largely not standardized. There have been efforts to bring rigor to circuit evaluations (Miller et al., 2024)—and to ascertain whether circuits are sensible data structures at all (Shi et al., 2024). As far as we are aware, no large-scale benchmark exists to systematically compare circuit localization methods. Thus, in addition to the proposed models and tasks (§2), we propose two metrics that enable comparison across circuit discovery *methods*, rather than across individual circuits.

A common metric for evaluating circuits is **faithfulness** (Wang et al., 2023; Miller et al., 2024). It aims to measure the extent to which a subgraph $\mathcal{C}$ of the computation graph explains the full model $\mathcal{N}$'s behavior on some task. Faithfulness is often defined ad hoc. In fact, this metric is often used for two distinct goals: (i) to measure whether $\mathcal{C}$ contributes to higher performance on a task (e.g., Meng et al., 2022; Stolfo et al., 2023; Nikankin et al., 2025), or (ii) to capture *any* component with measurable impact on a task, whether positive or negative (e.g., Wang et al., 2023; Hanna et al., 2024; Marks et al., 2025). Which is the correct way to measure the quality of a circuit? Many studies work with either one of the notions, or a mix of the two—e.g., discovering components as in (ii) but defining the metric more in line with (i), or vice versa. This overloads the term.

We claim that both are valid but complementary goals, and

therefore split faithfulness into (i) the **integrated circuit performance ratio** (CPR), and (ii) the **integrated circuit-model distance** (CMD). CPR prioritizes methods that locate components that improve model performance; higher is better. CMD prioritizes methods that locate components with *any* effect on model performance, including negative; 0 is best. We operationalize these below.

Discovering a circuit $\mathcal{C}$ often involves scoring components in a computation graph according to their importance, and only including those that exceed a causal importance threshold $\lambda$ (a hyperparameter; Conmy et al., 2023; Marks et al., 2025). Given $\mathcal{C}$ and the full neural network $\mathcal{N}$, we can define faithfulness $f$ as

$$f(\mathcal{C}, \mathcal{N}; m) = \frac{m(\mathcal{C}) - m(\varnothing)}{m(\mathcal{N}) - m(\varnothing)}, \qquad (1)$$

where $m$ is the logit difference $y' - y$ between the correct answer $y$ given the original input $x$ and correct answer $y'$ given the counterfactual input $x'$.[6] $\varnothing$ is the model with *all* components ablated (the empty circuit). Conceptually, this corresponds to the proportion of divergence in $m$ between the prior and the full model that the circuit recovers (Marks et al., 2025). There exist other formulations of $f$, like the *difference* (rather than *ratio*) of $m$ between $\mathcal{C}$ and $\mathcal{N}$ (Wang et al., 2023). We opt for the formulation of Marks et al. (2025), as this gives meaning to the values 0 ($\mathcal{C}$ recovers none of the performance of $\mathcal{N}$ relative to $\varnothing$) and 1 ($\mathcal{C}$ assigns identical probability differences to $y$ and $y'$ relative to $\mathcal{N}$).

We do not want the threshold $\lambda$ to affect comparisons between circuit localization methods. Thus, we propose shifting focus away from the quality of individual circuits, and toward a method's Pareto optimality with respect to (i) localizing task behavior, and (ii) minimizing circuit size.[7] Specifically, we quantify CPR as the area under the faithfulness curve w.r.t circuit size, and quantify CMD as the area between the faithfulness curve and 1. Both metrics evaluate faithfulness at many circuit sizes, and can be viewed as marginalizing over the circuit size hyperparameter.

In their exact form, CPR $= \int_{k=0}^{1} f(\mathcal{C}_k)\, dk$ and CMD $= \int_{k=0}^{1} |1 - f(\mathcal{C}_k)|\, dk$, where $f$ is the faithfulness of circuit $\mathcal{C}_k$ and $k$ is the proportion of edges from $\mathcal{N}$ in $\mathcal{C}_k$. We measure faithfulness at a few representative circuit sizes, and use these to approximate CPR and CMD:

1. For all proportions of components $k \in \{.001, .002, .005, .01, .02, .05, .1, .2, .5, 1\}$, discover a circuit $\mathcal{C}_k$

---

[6]No choice of $m$ is perfect. Like the counterfactual input, the metric defines the task. We follow Zhang & Nanda (2024), who recommend the logit difference based on empirical evidence.

[7]Prior implementations of minimality require manual circuit analysis (Wang et al., 2023); our formulation is more general, though it is useful primarily for relative comparisons, rather than as an absolute measure.

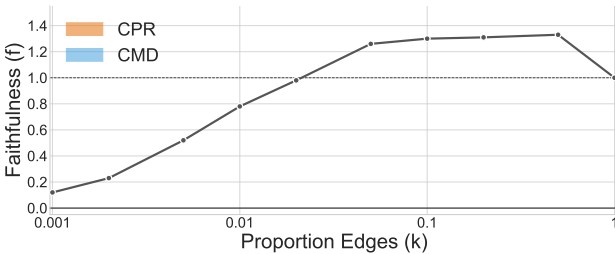

*Figure 2.* Definition of our faithfulness metrics. CPR, in orange, is the area under the faithfulness curve (the **black** line); it captures how well the method finds performant circuits at many circuit sizes. CMD, in blue, is the area between the faithfulness curve and the line at $f = 1$; it captures how closely the circuit's behavior resembles the model's task-specific behavior at many circuit sizes. Because we define $f$ as a ratio, $f = 1$ (the horizontal line) means that the circuit and full model achieve the same logit difference.

such that $\frac{|\mathcal{C}_k|}{|\mathcal{N}|} \leq k$.

2. Compute the faithfulness $f$ for all $\mathcal{C}_k$.
3. Compute the area under $f$ (CPR) and the area between $f$ and 1 (CMD) using the trapezoidal rule.

We illustrate CPR and CMD in Figure 2.

In a realistic neural network, it is difficult to anticipate the best-case and worst-case CPR or CMD values, meaning we cannot bound the metric without losing information. Thus, inspired by InterpBench (Gupta et al., 2024), we also train a model that implements a known ground-truth circuit for IOI. Because we know which edges are in the circuit, we report AUROC ($\in [0, 1]$) over edges. See App. F for InterpBench model training and implementation details.

**Measuring circuit size.** Circuits can be defined at many levels of granularity: entire layers, submodules in a layer, neurons in a submodule, etc. One can also define circuits at the level of nodes (e.g., submodules) or edges (e.g., connections between submodules). Thus, a key challenge is defining a notion of circuit size that allows comparing across different types of circuits. To this end, we treat including a node as equivalent to including all of its outgoing edges. Including $n$ neurons[8] (out of $d_{\text{model}}$) from submodule $u$ can be conceptualized as including all outgoing edges from $u$, scaled by a factor of $\frac{n}{d_{\text{model}}}$. Under these assumptions, we define the **weighted edge count**:

$$|\mathcal{C}| = \sum_{(u,v)\in\mathcal{C}} \left( \frac{|N_u \cap N_{\mathcal{C}}|}{|N_u|} \right), \qquad (2)$$

where $u$ and $v$ are nodes (submodules), $N_u$ is the set of neurons in $u$ (the size of which is typically $d_{\text{model}}$), and $N_{\mathcal{C}}$ is the set of neurons in the circuit. Intuitively, this is the num-

---

[8]We use "neuron" to refer to a single dimension of any hidden vector, regardless of whether it is preceded by a non-linearity.

ber of edges from a submodule weighted by the proportion of neurons from that submodule in the circuit; we sum this quantity over all submodules. We then normalize this count by the number of possible edges to obtain a percentage.

### 3.2. Circuit Localization Baselines

Here, we evaluate common circuit localization methods. We evaluate each model[9] and method[10] in §2 where possible. We compare across multiple axes of variation, including (a) circuit localization method (see below), (b) granularity, including edge-level and neuron-level circuits, and (c) ablation type, including counterfactual (CF) ablations, mean ablations, and "optimal ablations" (OA; Li & Janson, 2024). For all methods, we assign importance scores to all edges or nodes, and either include the top-scoring components or perform greedy search; see App. E.1 for details.

As a sanity check, we compare to random control circuits (RANDOM). We operationalize this by uniformly sampling an importance score in $[-1, 1]$ for all edges in the model. We take the mean CPR and CMD across 3 random seeds.

One way to find circuits is to filter model components by their indirect effects (IE; Pearl, 2001). The IE is defined as the change in $m$ caused by replacing a component's activation with its activation on another input, typically one where the expected output differs. We follow the procedure of Vig et al. (2020) and (Finlayson et al., 2021). However, computing IE in exact form (ACTIVATION PATCHING, or ActP) is expensive, requiring $O(n)$ forward passes, where $n$ is the number of possible edges in $\mathcal{N}$.

**Attribution methods** aim to reduce the cost of computing IE by approximating it. Nanda (2023) and Syed et al. (2024) propose ATTRIBUTION PATCHING (AP), which linearly approximates the IE for all nodes or edges in $O(1)$ forward passes. When performed at the node level, we call this NODE AP (NAP); at the edge level, it is EDGE AP (EAP).

Unfortunately, AP approximates IE poorly (Syed et al., 2024). AP WITH INTEGRATED GRADIENTS (AP-IG; Sundararajan et al. 2017) improves on AP by performing multiple steps of AP, trading off speed for approximation quality. We test two AP-IG variants: (i) AP-IG-inputs (Hanna et al., 2024) and (ii) AP-IG-activations (Marks et al., 2025). AP-IG-inputs requires $O(Z)$ forward passes, while AP-IG-activations requires $O(Z \cdot L)$, where $Z$ is the number of AP steps, and $L$ is the number of model layers. We use $Z = 5$, following Hanna et al. (2024). We explore AP-IG at the node level (NAP-IG) and edge level (EAP-IG). See

---

[9]We do not evaluate Gemma or Qwen on Arithmetic, as they tokenize numbers such that each digit has its own token.

[10]Exact activation patching and optimal ablations become intractable with respect to runtime as models scale. UGS becomes intractable with respect to memory requirements as models scale.

*Table 2.* CMD scores across circuit localization methods and ablation types (lower is better), and AUROC scores for InterpBench (higher is better). All evaluations were performed using counterfactual ablations. Arithmetic scores are averaged across addition and subtraction; see Table 17 (App. E.3) for separate scores. We **bold** and underline the best and second-best methods per column, respectively.

| Method | InterpBench (↑) | IOI | | | | Arithmetic | MCQA | | | ARC (E) | | ARC (C) |
| | | GPT-2 | Qwen-2.5 | Gemma-2 | Llama-3.1 | Llama-3.1 | Qwen-2.5 | Gemma-2 | Llama-3.1 | Gemma-2 | Llama-3.1 | Llama-3.1 |
| --- | --- | --- | --- | --- | --- | --- | --- | --- | --- | --- | --- | --- |
| Random | 0.44 | 0.75 | 0.72 | 0.69 | 0.74 | 0.75 | 0.73 | 0.68 | 0.74 | 0.68 | 0.74 | 0.74 |
| EActP (CF) | 0.28 | **0.02** | 0.49 | - | - | - | 0.36 | - | - | - | - | - |
| EAP (mean) | 0.78 | 0.29 | 0.18 | 0.25 | 0.04 | 0.07 | 0.21 | 0.20 | 0.16 | 0.22 | 0.28 | 0.20 |
| EAP (CF) | 0.73 | 0.03 | 0.15 | 0.06 | **0.01** | 0.01 | 0.07 | 0.08 | **0.09** | **0.04** | **0.11** | **0.18** |
| EAP (OA) | 0.77 | 0.30 | 0.16 | - | - | - | 0.11 | - | - | - | - | - |
| EAP-IG-inp. (CF) | 0.71 | 0.03 | 0.02 | 0.04 | **0.01** | **0.00** | 0.08 | **0.06** | 0.14 | **0.04** | **0.11** | 0.22 |
| EAP-IG-act. (CF) | **0.81** | 0.03 | **0.01** | **0.03** | **0.01** | **0.00** | **0.05** | 0.07 | 0.13 | **0.04** | 0.30 | 0.37 |
| NAP (CF) | 0.30 | 0.38 | 0.33 | 0.37 | 0.29 | 0.28 | 0.30 | 0.35 | 0.32 | 0.33 | 0.69 | 0.69 |
| NAP-IG (CF) | 0.62 | 0.27 | 0.20 | 0.26 | 0.19 | 0.18 | 0.18 | 0.29 | 0.33 | 0.28 | 0.67 | 0.67 |
| IFR | 0.71 | 0.42 | 0.69 | 0.75 | 0.83 | 0.22 | 0.60 | 0.62 | 0.48 | 0.66 | 0.64 | 0.76 |
| UGS | 0.74 | 0.03 | 0.03 | - | - | - | 0.20 | - | - | - | - | - |

App. E.2 for definitions and details.

**Information flow routes** (IFR; Ferrando & Voita, 2024) is a non-counterfactual-based and non-causal method that includes edge $(u, v)$ in $\mathcal{C}$ if the output vector of $u$ and input vector of $v$ are highly similar; this is taken as evidence of a writing operation. See App. E.2 for details on how we adapted IFR to our formalization of circuits.

**Mask-based methods** aim to learn a pruning mask on the edges of the computational graph. During training, masks are continuous. Edges with low mask values are ablated more often or more fully; edges with high values are likely in the circuit. The mask's training objective aims to maintain model behavior (often measured by KL-divergence with the unablated model) while keeping the mask sparse (measured by its $L_1$ norm). After training, the mask can be converted into a binary mask indicating which edges are in the circuit.

As a mask-based method, we employ UNIFORM GRADIENT SAMPLING (UGS; Li & Janson, 2024), which uses CF ablations. Li & Janson also propose optimal ablations (OA), in which the mask is learned jointly with the value used to ablate non-circuit edges; they prefer OA as it is both independent of the example being ablated (unlike CF ablations) and minimally harmful (unlike mean ablations). Due to computational constraints, we do not do this, but instead learn OA vectors by optimizing them to minimize the expected cross-entropy loss on the task dataset. We then use these vectors as ablation values when running circuit localization with EAP. See App. E.2 for details.

### 3.3. Results

We present CMD and AUROC scores (Table 2) for each method and task where it is tractable to run; see Table 14 in App. E for CPR scores. On both metrics, EAP-IG-inputs with CF ablations generally performs best. EAP-IG-activations and UGS are competitive with EAP-IG-inputs

w.r.t. CMD, but have higher runtime and memory usage respectively. For CPR, they are less competitive; this is unsurprising for UGS, as it directly optimizes for maintaining model behavior, rather than finding performant components.

More surprisingly, edge activation patching (EActP) does not always perform best, despite computing *exact* IEs for each edge: it dominates for IOI on GPT-2, but not Qwen-2.5 or InterpBench. This could stem from our use of fewer examples to run EActP due to its long runtime. But EActP also has deeper limitations: like attribution methods, it estimates the effect of ablating each edge independently; this may imperfectly predict the effect of ablating multiple edges in tandem (Mueller, 2024). That said, UGS, which considers many edges at once, is also not the top performer.

Circuits found with CF ablations outperform those found with mean or optimal ablations; the latter two score similarly to each other. This is expected, as CF ablations more closely resemble the evaluation setting, and more precisely localize the distinction captured by the CF input pairs.

Certain methods underperform on all metrics: node-level circuits do poorly, likely because each node "costs" many edges—they are not as sparse as edge circuits. IFR achieves lower performance than attribution methods, but still significantly better than random circuits.

In summary, **EAP-IG-inputs achieves the highest performance** on average on both CMD and CPR. However, **other techniques**, like EAP-IG-activations, EAP, and UGS, **remain competitive**.

## 4. Causal Variable Localization Track

The circuit localization track evaluates methods that localize *behaviors* to model components that form *end-to-end pathways* from input to output. In contrast, the causal variable localization track evaluates methods that localize specific

*concepts* along causally active paths. Figure 6 (App. G) illustrates an example evaluation in this track, while Table 3 contains results for causal variable localization methods.

## 4.1. Causal Abstraction

A basic assumption in mechanistic interpretability is that models implement intelligent behaviors by representing and manipulating concepts.[11] We operationalize such hypotheses by encoding the reasoning process as a causal model $\mathcal{H}$ with variables corresponding to concepts. The task is to align these high-level conceptual variables in a causal model with low-level features in a neural model that have the same mechanistic role, i.e., the high-level causal model is a *causal abstraction* of the LM (Geiger et al., 2021; 2024a).

**LM Features.** What should be the atomic units of analysis, or *features*, for mechanistic interpretability? The answer is hotly debated and not currently clear (Mueller et al., 2024), so we design this track to incentivize investigation of this fundamental question. We adopt the framework of Geiger et al. (2024a) in which any hidden vector $\mathbf{h} \in \mathbb{R}^d$ constructed by a model $\mathcal{N}$ during inference can be mapped into a new feature space $\mathbb{F}^k$ (e.g., a rotated vector space) using an invertible function $\mathcal{F}: \mathbb{R}^d \to \mathbb{F}^k$ (e.g., multiplication with an orthogonal matrix). Features $\Pi$ are a set of indices between 1 and $k$, i.e., a set of dimensions in $\mathbb{F}^k$. This framework supports a variety of features, including neurons, orthogonal directions, SAE features, and non-linear features. The vector $\mathbf{h}$ might come from the residual stream between transformer layers or the output of an attention head.

**Alignments.** Alignments between high-level conceptual variables and low-level features will not be static, even in the simplest cases. For instance, in MCQA, the index of the token corresponding to the correct multiple choice answer will change depending on the number of tokens in the question. As such, submissions to MIB can provide an **alignment** that aligns a variable $X$ with features $\Pi_X$ of a dynamically selected hidden vector $\mathbf{h}$, e.g., the residual stream of the correct answer token in the MCQA task.

**Faithfulness metrics.** We quantify the degree to which a variable $X \in \mathcal{H}$ faithfully abstracts the features $\Pi_X$ using *interchange interventions*. Given base and counterfactual inputs $(b, c)$, the interchange intervention $\mathcal{H}_{X \leftarrow \mathsf{Get}(\mathcal{H}(c), X)}(b)$ runs $\mathcal{H}$ on base input $b$ while fixing the variable $X$ to the value it takes when $\mathcal{H}$ is run on a counterfactual input $c$ (Vig et al., 2020; Geiger et al., 2020). The distributed interchange intervention $\mathcal{N}_{\Pi_X \leftarrow \mathsf{Get}(\mathcal{N}(c), \Pi_X)}(b)$ runs $\mathcal{N}$ on $b$ while fixing the features $\Pi_X$ of the hidden vector $\mathbf{h}$ passed through $\mathcal{F}$ to the value they take for counterfactual input $c$ (Wu et al., 2023; Amini et al., 2023; Geiger et al., 2024b).

---

[11]Crucially, the concepts employed by an LM may not relate to those employed by a human on the same task (Hewitt et al., 2025).

Given a counterfactual dataset $\mathcal{D}$ and a high-level causal model $\mathcal{H}$ aligned to an LM $\mathcal{N}$, we measure whether interchange interventions on a variable $X$ in $\mathcal{H}$ and aligned features $\Pi_X$ in $\mathcal{N}$ produce the same output:

$$\mathsf{Faith}(X, \Pi_X, \mathcal{H}, \mathcal{D}) =$$
$$\sum_{(b,c) \in \mathcal{D}} \left[ \mathcal{H}_{X \leftarrow \mathsf{Get}(\mathcal{H}(c), X)}(b) = \mathcal{N}_{\Pi_X \leftarrow \mathsf{Get}(\mathcal{N}(c), \Pi_X)}(b) \right].$$

This metric of *interchange intervention accuracy* (IIA) is for all tasks except IOI, which has a logit-based metric (§4.4).

For each task, we have base and counterfactual input pairs (§2.1). We use interchange interventions on these pairs to isolate variables in $\mathcal{H}$; interventions to a variable should result in predictable changes to a model's outputs. This is not as obvious as it first appears; naïve approaches to sampling counterfactual inputs used for intervention can undersample or exclude crucial settings. We filter out all examples where the model predicts the incorrect output for the base input or any of the counterfactuals used.

**Featurizers and feature selection.** We consider five baselines for constructing the featurizer $\mathcal{F}$ and selecting the features $\Pi_X$. See App. G for further details.

We evaluate three *unsupervised* featurization methods that provide features without access to a high-level causal model. The most naïve one is the "Full Vector" baseline; this entails using an identity featurizer and selecting all features—i.e., intervening on the full untransformed hidden vector $\mathbf{h}$. We also evaluate PRINCIPAL COMPONENT ANALYSIS (PCA; Tigges et al. 2023; Marks & Tegmark 2024) and SPARSE AUTOENCODERS (SAE; Bricken et al. 2023; Huben et al. 2024), which encode into very high-dimensional spaces with many features. For SAEs, we use GemmaScope (Lieberum et al., 2024) and LlamaScope (He et al., 2024).

To select SAE features, principal components, or standard dimensions of hidden vectors that are aligned with high-level causal variables, we use DESIDERATA-BASED MASKING (DBM; Cao et al. 2020; 2022; Csordás et al. 2021; Davies et al. 2023; Chaudhary & Geiger 2024) to learn a binary mask over features using a high-level causal model as a source of supervision. The masks are trained to maximize the faithfulness metric on training data.

We also evaluate a *supervised* featurization method DISTRIBUTED ALIGNMENT SEARCH (DAS; Geiger et al. 2024b) that learns a featurizer with supervision from the high-level causal model. First, a variable in the high-level causal model is aligned with features that are randomly initialized orthogonal directions that define a linear subspace of a hidden vector. Then, the features are trained to maximize the faithfulness metric. There is no need for a separate feature selection procedure because the features are constructed specifically for the high-level variable.

For tasks other than IOI, we brute-force search over a few manually selected token locations. For each layer and token location, we search for features of the residual stream vector corresponding to variables in the high-level causal model. For each causal variable, we use training data to create and select features, and then evaluate the faithfulness of aligning those features with the causal variable. We take the token location with the highest score at each layer, and report best layer and average across layers. Future submissions may target any token(s). For IOI, we focus our experiments on the attention heads identified by Wang et al. (2023) in GPT2-small, but we will allow for future submissions to identify new attention heads in the other three models.

## 4.2. MCQA and ARC (Easy)

**Causal model.** For the two multiple-choice datasets, we hypothesize LMs compute the position of the answer token in context before retrieving the answer token itself. $\mathcal{H}_{\mathbf{MCQA}}$ is an algorithm with three variables: a text input variable $T$, an *ordering ID* $X_{\mathrm{Order}}$ (Dai et al., 2024) storing the position of the answer, and the answer token $O_{\mathrm{Answer}}$. This model abstracts away the details of how the answer position is computed; the mechanism for $X_{\mathrm{Order}}$ is a lookup table from inputs to the index of the answer token, i.e., a number between 1 and 4 because there are four choices. Instead, the focus is on the retrieval of the correct choice; the mechanism for $O_{\mathrm{Answer}}$ dereferences the index stored in $X_{\mathrm{Order}}$.

**Counterfactuals.** We use counterfactuals where the answer position is changed, where the choice letters are randomized, or both. When both the answer position and the choice letters are different in the counterfactual, a different output is expected when localizing the ordering id $X_{\mathrm{Order}}$ versus the answer token $O_{\mathrm{Answer}}$. If the ordering ID is targeted, then the expected output is the choice token in the base at the answer position from the counterfactual. If the answer token is targeted, then the expected output is the answer token from the counterfactual, regardless of position.

**Results.** We target the residual stream of the last token of the input and the correct choice letter token at each layer. We generally see strong evidence (Tables 3a and 3c; App. G.5.1) of the causal model $\mathcal{H}_{\mathbf{MCQA}}$ being a faithful abstraction, with DAS successfully disentangling the ordering ID variable $X_{\mathrm{Order}}$ from the output token $O_{\mathrm{Answer}}$ in many layers. Even the full vector baseline successfully localizes the variable in some layers, though it performs poorly on average because both variables are aligned with the same features.

## 4.3. Two-Digit Addition

**Causal model.** For the two-digit addition task, we hypothesize that LMs use a "carry-the-one" algorithm, as illustrated in Figure 6. The causal model $\mathcal{H}_+$ has a text input variable $T$ that is parsed into the variables $X_1$, $X_{10}$, $Y_1$, and $Y_{10}$,

representing the ones and tens digits of each two-digit input. The variable $X_{\mathrm{Carry}}$ is a child of $X_1$ and $Y_1$ and takes value 1 if $X_1 + Y_1 > 10$. The output variable $O_{110}$ has all inputs and $X_{\mathrm{Carry}}$ as parents and takes on the value $X_{\mathrm{Carry}} + X_{10} + Y_{10}$. The output variable $O_1$ has $X_1$ and $Y_1$ as parents and takes on the value $(X_1 + Y_1)\%10$. For the benchmark, we report results for localizing the variable $X_{\mathrm{Carry}}$.

**Counterfactual dataset.** We use equal parts random counterfactuals and counterfactuals that do not change the carry-the-one variable (e.g., base *17+75* and counterfactual *11+71*). Interchange interventions on $X_{\mathrm{Carry}}$ in $\mathcal{H}_+$ with random counterfactuals will cause the output variable $O_{110}$ to increase or decrease by 1 half the time and have no effect the other half. The carry-the-one counterfactual inputs always require a change in the output, but hold the input and parts of the output fixed so that low-level interchange interventions are less likely to have unintended consequences.

**Results.** For each baseline, we target the last token and the last token of the second operand. We see poor performance across the board (Table 3b; App. G.5.2). The Gemma-2 and Llama-3.1 results show only faint signs of success. There may be no "carry-the-one" variable present in these models. Alternatively, the variable might be represented in a *nonlinear features space*, e.g., an onion representation (Csordás et al., 2024). Future submissions that beat baselines will require genuine progress.

## 4.4. Indirect Object Identification

App. A of Wang et al. (2023) describes an experiment where the four "S-Inhibition" heads that decrease the probability of the subject token are intervened upon to invert the subject's position or token identity. The heads contain token and position signals that make roughly linear contributions to the difference between the indirect object and subject logits. However, they did not try to disentangle the hidden vector outputs of those heads to align each signal with LM features.

**Causal model.** First, we replicate these experiments on our datasets and fit a linear model; we use this to define a causal model $\mathcal{H}_{\mathbf{IOI}}$ that predicts the logit difference between the indirect object and subject. $\mathcal{H}_{\mathbf{IOI}}$ takes a text input $T$ and computes the token and positional information $S_{\mathrm{Tok}}$ and $S_{\mathrm{Pos}}$ of the subject. Then, the output variable $O$: (i) checks if the token and position variables $S_{\mathrm{Tok}}$ and $S_{\mathrm{Pos}}$ match the input $T$ and inverts the signal if a mismatch is detected, and (ii) computes the logit difference between the indirect object and subject as a linear function of these binary variables.

**Counterfactuals.** We use counterfactuals where the subject and indirect object tokens are inverted, their position is inverted, or both. We align each variable with features of the four heads. These counterfactuals test the ability of methods to disentangle the token and position variables.

*Table 3.* Baseline results for the causal variable localization track. In each table, the first row is the task, the second row is the model, and the third row is the causal variable. For Arithmetic, MCQA, and ARC (Easy), we report interchange intervention accuracy, i.e., the proportion of aligned interventions on the causal model and deep learning model that result in the same output token(s); higher is better. For each method of aligning a causal variable to LM features, we report the mean across counterfactual datasets and layers in the low-level model. In parenthesis and bold, we report the best alignment across all layers. For IOI, we report the mean-squared error between the causal model logit and the deep learning model logit; lower is better. See App. G.5 for more detailed results by task.

| | ARC (Easy) | | | |
| | Gemma-2 | | Llama-3.1 | |
| Method | $O_{\text{Answer}}$ | $X_{\text{Order}}$ | $O_{\text{Answer}}$ | $X_{\text{Order}}$ |
|---|---|---|---|---|
| DAS | 88 (**94**) | 76 (**88**) | 88 (**99**) | 74 (**84**) |
| DBM | 82 (**99**) | 63 (**80**) | 85 (**100**) | 69 (**82**) |
| +PCA | 78 (**98**) | 64 (**81**) | 84 (**100**) | 72 (**83**) |
| +SAE | 70 (**89**) | 54 (**70**) | 74 (**94**) | 55 (**67**) |
| Full Vector | 63 (**100**) | 43 (**74**) | 68 (**100**) | 47 (**72**) |

(a) The ARC (Easy) task with a high-level model that computes the ordering of the answer $X_{\text{Order}}$ and then the answer token $O_{\text{Answer}}$.

| | Arithmetic (+) | |
| | Gemma-2 | Llama-3.1 |
| Method | $X_{\text{Carry}}$ | $X_{\text{Carry}}$ |
|---|---|---|
| DAS | 56 (**69**) | 54 (**65**) |
| DBM | 42 (**56**) | 47 (**58**) |
| +PCA | 39 (**56**) | 37 (**56**) |
| +SAE | 42 (**56**) | 38 (**55**) |
| Full Vector | 27 (**34**) | 35 (**45**) |

(b) The two-digit arithmetic task with a variable that computes the carry-the-one variable $X_{\text{Carry}}$.

| | MCQA | | | | | |
| | Gemma-2 | | Llama-3.1 | | Qwen-2.5 | |
| Method | $O_{\text{Answer}}$ | $X_{\text{Order}}$ | $O_{\text{Answer}}$ | $X_{\text{Order}}$ | $O_{\text{Answer}}$ | $X_{\text{Order}}$ |
|---|---|---|---|---|---|---|
| DAS | 95 (**97**) | 77 (**93**) | 94 (**100**) | 77 (**91**) | 86 (**95**) | 78 (**100**) |
| DBM | 84 (**99**) | 63 (**84**) | 86 (**100**) | 66 (**73**) | 46 (**94**) | 60 (**99**) |
| +PCA | 57 (**96**) | 52 (**81**) | 65 (**99**) | 53 (**74**) | 22 (**76**) | 54 (**100**) |
| +SAE | 73 (**90**) | 51 (**65**) | 80 (**99**) | 58 (**65**) | – | – |
| Full Vector | 61 (**100**) | 44 (**77**) | 77 (**100**) | 46 (**68**) | 35 (**99**) | 49 (**99**) |

(c) The MCQA task with variables for the ordering of the answer $X_{\text{Order}}$ and then the answer token $O_{\text{Answer}}$. This is a low-data regime ($\approx$100 examples).

| | IOI | |
| | GPT-2 | |
| Method | $S_{\text{Pos}}$ | $S_{\text{Tok}}$ |
|---|---|---|
| DAS | 2.20 | 2.08 |
| DBM | 2.22 | 2.35 |
| +PCA | 2.24 | 2.33 |
| Full Vector | 2.45 | 2.82 |

(d) The IOI task with variables for the position $S_{\text{Pos}}$ and token $S_{\text{Tok}}$ of the subject. The metric is mean-squared error; lower is better.

**Results.** Broadly, we find evidence that the position and token variable can be disentangled (Table 3d; App. G.5.4). For the full vector baseline, we conduct a brute force search and find an alignment of position to heads 7.3, 7.9 ($\Pi_{S_{\text{Pos}}}$), and 8.6 and token to head 8.10 ($\Pi_{S_{\text{Tok}}}$), to be better than other alignments to entire heads. While the variables can be disentangled at the level of heads, even better results are achieved when each variable is aligned to features of heads.

### 4.5. General Discussion

**Distributed alignment search (DAS) consistently achieves the best results.** DAS is the only method that learns features with supervision from the high-level causal model, so this is not surprising. **DBM on standard hidden dimensions is often successful**; This shows that the dimensions of untransformed hidden vectors can be useful units of analysis; nonetheless, DAS outperforming DBM shows that non-basis-aligned directions in activation space are generally better units of analysis than basis-aligned directions.

**DBM on PCA or SAE features is not better than DBM on standard dimensions.** SAE and PCA features generally fail to improve upon neurons, i.e., standard dimensions of hidden vectors, as a unit of analysis. The especially poor performance of PCA on MCQA may be due to the low-data regime of $\approx$100 examples. This is in line with results from AxBench, where SAEs struggle against simple steering baselines (Wu et al., 2025).

**We include an additional task on disentangling factual knowledge from the RAVEL benchmark (Huang et al., 2024a).** See Appendix G.5.3 for baseline results and details.

**There is room for future submissions to establish state-of-the-art results on the benchmark.** First, our baselines provide weak evidence for the carry-the-one variable in the addition task. Second, we only run baselines for IOI on the GPT-2 attention heads identified by (Wang et al., 2023); we encourage future submissions to locate and featurize attention heads in the other three models. Third, we did not conduct exhaustive hyperparameter searches for any of the baseline methods, so there is likely room for improvement.

## 5. Conclusion

We have proposed MIB, a Mechanistic Interpretability Benchmark, and demonstrated its value for directly comparing mechanistic interpretability methods. MIB corroborates recent findings, like the value of attribution methods, and challenges others, like the utility of SAEs as featurizers for known causal variables. MIB is not in its final form: as progress in MI is rapid, we intend this as a **living benchmark** that scales to incorporate new advances in the field.

## Acknowledgments

We are grateful for feedback and ideas discussed with Neel Nanda, Sandro Pezzelle, Christopher Potts, and Tamar Rott Shaham in an early phase of this project.

This research was supported by a postdoctoral fellowship under the Zuckerman STEM Leadership Program (A.M.), grants from Open Philanthropy (A.G., Y.B., N.P., D.B.), grants from AI Safety Support Ltd (I.A., R.G.), the OpenAI Superalignment Fast Grant Program (M.H.), the Ariane de Rothschild Women Doctoral Program (D.A.), an armasuisse CYD Doctoral Fellowship (Al.St.), an Azrieli International Postdoctoral Fellowship (A.R.), an Azrieli Foundation Early Career Faculty Fellowship (Y.B.), a Google academic gift (Y.B.), the European Union (ERC, Control-LM, 101165402; Y.B.), and the Apple AIML PhD fellowship (H.O.). The opinions expressed here are those of the authors, and not of any of the mentioned organizations.

## Impact Statement

This paper presents work whose goal is to standardize the evaluation of mechanistic interpretability methods. Advancements in interpretability methods will advance current approaches to AI safety and robustness, many of which rely on localization as part of their pipelines. This could also potentially result in better countermeasures against safety methods that are meant to increase harm. We do not anticipate that MIB will directly contribute to such harms; it will primarily allow researchers to come to stronger conclusions regarding which localization methods tend to be best *in general*, regardless of downstream application.

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

## A. Table of Notation

In Table 4, we summarize the mathematical notation used throughout the paper, grouped by the track(s) they appear in.

## B. Related Work

**Circuit discovery evaluation.** While there do not exist benchmarks for circuit discovery methods in general, there do exist targeted tests of whether the *concept of* circuits is sensible (Shi et al., 2024). There are also benchmarks designed with ground-truth circuits in mind (Gupta et al., 2024), though the tasks in this benchmark are relatively simple, as the circuits are hand-crafted. Methods and metrics papers tend to focus on only one or two tasks and only one or two models (Miller et al., 2024; Ferrando & Voita, 2024; Conmy et al., 2023); these can function as strong proofs of concept, but limit our understanding of the generalizability and scalability of these methods and metrics.

**Causal variable localization evaluations.** The RAVEL (Huang et al., 2024a) and CausalGym benchmarks (Arora et al., 2024) both enable comparisons across featurization methods, though in more narrow domains. The SAEBench (Karvonen et al., 2025) is similar in concept, though much narrower in scope w.r.t. the kinds of methods that can be evaluated (i.e., only SAEs). Our benchmark compares across a range of tasks, models, and methods.

**Other evaluations.** There exist benchmarks that do not fall cleanly within these two camps. An impactful application of MI is targeted model editing (Meng et al., 2022), for which there now exist multiple benchmarks (Cohen et al., 2024; Abraham et al., 2022; Zhong et al., 2023). An emerging paradigm is evaluating automated *interpretability agents*; for example, FIND (Schwettmann et al., 2023) evaluates whether interpretability agents correctly describe latent functions implemented by model components. Other benchmarks focus on benchmarking explanations of LM behaviors (Mills et al., 2023; Atanasova et al., 2023).

## C. Limitations

Our two tracks separate the problems of featurization and causal dependency location for cleaner evaluation. However, these are mutually influential problems: one could potentially locate better circuits by first decomposing MLPs into sparse features, for example (Marks et al., 2025). The featurization method one uses should also be informed by the downstream task; for example, the outputs of DAS are not immediately applicable to finding circuits. Future work should consider the joint problem of (1) building causal dependency graphs from (2) more meaningful units, where these units may potentially exist at various levels of granularity.

For circuit localization, there exist metrics such as completeness that cannot be tractably computed without access to the ground-truth set of causally relevant components. This motivated our inclusion of the InterpBench model, whose AUROC metric includes completeness. Some work has attempted to measure completeness as the faithfulness of the circuit's complement (Marks et al., 2025), but as it is easy to reduce performance even without ablating the full circuit, this may not be a good signal for when one has recovered the full set of causally relevant dependencies. We acknowledge that a fully automated completeness score is absent for the remaining models.

For causal variable localization, our faithfulness metric captures the extent to which the causal variable—not the entire high-level model—aligns with the representation. The high-level model may differ from than the hypothesis, but it would still be possible to modify the model's behavior in a predictable way, and we believe this will be reflected in the scores. Nonetheless, this paradigm presumes the existence of the high-level model in the computation graph. We have mainly included graphs for which there exists evidence from past work, but we acknowledge that these graphs may not always exist in the models we evaluate in the exact forms shown here.

Finally, our benchmark focuses solely on large language models. Given that this is the current focus of the vast majority of mechanistic interpretability research, we believe that this gives a broad-coverage sample of models commonly studied in the literature. It would be helpful in future work to expand the scope of these evaluations to include other modalities.

## D. Further Details on Materials

Table 5 contains dataset statistics.

*Table 4.* The notation used throughout the paper, grouped by track.

| Track | Symbol | Meaning |
|---|---|---|
| Shared | $\mathcal{N}$ | The full computation graph; a neural network |
| | $d_{\text{model}}$ | The size of (i.e., number of neurons in) the output vector of each layer |
| Circuit Localization | $\mathcal{C}$ | A circuit $\in \mathcal{N}$ |
| | $k$ | The proportion of edges in a circuit. If a circuit has $\leq k \times 100$ edges, this is sometimes expressed as $\mathcal{C}_k$ |
| | $u$ | A node in the computation graph. Could be an MLP or an attention head |
| | $(u, v)$ | An edge from node $u$ to node $v$ |
| | $m$ | The metric used to evaluate a circuit. Usually the logit difference between a correct and incorrect answer |
| | $f$ | The faithfulness of a single circuit. Defined as a ratio of $m$ given $\mathcal{C}$ and $m$ given $\mathcal{N}$ |
| | CPR | Circuit performance ratio (higher is better). Defined as the area under the faithfulness curve. An aggregation over $f$ values at many circuit sizes |
| | CMD | Circuit-model distance (0 is best). Area between the faithfulness curve and 1. An aggregation over $f$ values at many circuit sizes |
| | $N_u$ | The number of neurons in node $u$, usually equal to $d_{\text{model}}$ |
| | $N_{\mathcal{C}}$ | The set of all neurons in $\mathcal{C}$ |
| Causal Variable Localization | $\mathcal{H}$ | A high-level causal model |
| | $X$ | A variable in $\mathcal{H}$ |
| | $b$ | An base input to the neural network |
| | $c$ | An counterfactual input to the neural network that differs from $b$ in some systematic way |
| | $\mathbf{h}$ | The output activation vector of a node in the computation graph |
| | $\mathcal{F}$ | The featurization function that transforms $\mathbf{h}$ to a new space where the causal variable $X$ is easier to isolate |
| | $\Pi_X$ | A set of "features" in a hidden vector $\mathbf{h}$ abstracted by a variable $X$, i.e., dimensions in the range of $\mathcal{F}$ that encode $X$ |
| | $\mathcal{D}$ | A dataset containing $(b, c)$ pairs |
| | $\mathcal{H}_{X \leftarrow \text{get}(\mathcal{H}(c), X)}(b)$ | An interchange intervention on the high-level model $\mathcal{H}$ which is run on the input $b$ while the variable $X$ is fixed to the value it takes when $\mathcal{H}$ is run on input $c$. |
| | $\mathcal{N}_{\Pi_X \leftarrow \text{get}(\mathcal{N}(c), \Pi_X)}(b)$ | A distributed interchange intervention on the LM $\mathcal{N}$ which is run on the input $b$ while the features $\Pi_X$ of a hidden vector $\mathbf{h}$ are fixed to the value they take when $\mathcal{N}$ is run on input $c$. |

*Table 5.* Information about datasets and their splits.

| Dataset | Train | Validation | Test (Public/Private) |
|---|---|---|---|
| IOI | 10000 | 10000 | 1000/1000 |
| MCQA | 110 | 50 | 50/50 |
| Arithmetic (+) | 34400 | 4920 | 1000/1000 |
| Arithmetic (−) | 17400 | 2484 | 1000/1000 |
| ARC (Easy) | 2251 | 570 | 1188/1188 |
| ARC (Challenge) | 1119 | 299 | 586/586 |
| RAVEL | 100000 | 16000 | 1000 |

### D.1. Indirect Object Identification (IOI)

To generate IOI examples, we collect sets of templates and attributes—namely, common English first names, common place names, and everyday objects. We separate the templates and attributes into four disjoint groups and use them to generate the four splits (public train/validation/test and private test sets). This means that different splits do not share any attributes. We generate 10,000 IOI examples per split using 43 templates, 166 first names, 319 object names, and 247 place names. The templates and attributes used in the public sets partly overlap with the original dataset by Wang et al. (2023). The rest of the public attributes and the additional private attributes were generated using ChatGPT (OpenAI, 2022) and manually verified. We verify that all names are tokenized to a single token using the test prompt *"I am {name}"* across our models.

In Figure 3, we provide an example from our IOI dataset. Each example includes a prompt, a prompt template, metadata, a list of choices, and the index of the correct completion (answer key) from the list of choices.

**Counterfactuals.** For each instance, we create eight counterfactuals: the six counterfactuals described by Wang et al. (2023), an additional counterfactual which is the composition of all three transformations proposed by Wang et al. (2023), and a counterfactual where the second instance of the subject is replaced with a third random name that did not appear in the first clause of the sentence ("ABC"). Table 6 contains an example of each counterfactual type.

In the circuit localization track, we use the IO ↔ S2 Flip counterfactual. In the causal variable track, we use the IO ↔ S1 Flip, IO ↔ S2 Flip, and IO ↔ S1 Flip + IO ↔ S2 Flip counterfactuals.

```
"prompt": "After Nick and John spent some time
at the car dealership, Nick offered a nail to",
"template": "After {name_A} and {name_B} spent some time
at the {place}, {name_C} offered a {object} to",
"metadata": {
  "indirect_object": "John",
  "subject": "Nick",
  "object": "nail",
  "place": "car dealership",
  "random_a": "Max",
  "random_b": "Fred",
  "random_c": "Bob"
},
"choices": [
  "John",
  "Nick"
],
"answerKey": 0,
"abc_counterfactual": {
  "prompt": "After Nick and John spent some time
  at the car dealership, Bob offered a nail to",
  "choices": [
    "John",
    "Nick",
    "Bob"
  ],
  "answerKey": -1
},
"random_names_counterfactual": {
  "prompt": "After Max and Fred spent some time
  at the car dealership, Max offered a nail to",
  "choices": [
    "Max",
    "Fred"
  ],
  "answerKey": 1
},
...
```

*Figure 3.* An IOI example. Each input is paired with a set of templatically generated counterfactuals.

*Table 6.* An IOI example and its 8 associated counterfactuals.

| Prompt / Counterfactual | Name A | Name B | Name C | Text | Correct Completion |
|---|---|---|---|---|---|
| Original Prompt | Nick | John | Nick | *After Nick and John spent some time at the car dealership, Nick offered a nail to* | John |
| ABC | Nick | John | Bob | *After Nick and John spent some time at the car dealership, Bob offered a nail to* | N/A |
| Random Names | Max | Fred | Max | *After Max and Fred spent some time at the car dealership, Max offered a nail to* | Fred |
| IO ↔ S1 Flip | John | Nick | Nick | *After John and Nick spent some time at the car dealership, Nick offered a nail to* | John |
| IO ↔ S2 Flip | Nick | John | John | *After Nick and John spent some time at the car dealership, John offered a nail to* | Nick |
| Random Names + IO ↔ S1 Flip | Fred | Max | Max | *After Fred and Max spent some time at the car dealership, Max offered a nail to* | Fred |
| Random Names + IO ↔ S2 Flip | Max | Fred | Fred | *After Max and Fred spent some time at the car dealership, Fred offered a nail to* | Max |
| IO ↔ S1 Flip + IO ↔ S2 Flip | John | Nick | John | *After John and Nick spent some time at the car dealership, John offered a nail to* | Nick |
| Random Names + IO ↔ S1 Flip + IO ↔ S2 Flip | Fred | Max | Fred | *After Fred and Max spent some time at the car dealership, Fred offered a nail to* | Max |

## D.2. Arithmetic

We list the templates used to format the arithmetic queries in Table 7. We consider four text-based prompts and two Arabic numeral-based prompts. The prompts are modified from Stolfo et al. (2023). An example instance and its associated counterfactuals are in Table 8. Subtraction queries are constrained to cases with positive results to maintain single-token answers when possible. Of the four models we investigate, two (Llama and GPT-2) tokenize numeric answers as single tokens and two (Qwen and Gemma) tokenize numbers into their respective digits (meaning that correct answers will often be more than one token in length). In the latter case, we generate the oracle number of tokens corresponding to the number of digits in the correct answer and then check for exact-match correctness.

**Counterfactuals.** We create counterfactuals by adjusting the operands in ways that will affect not only the correct answer, but also the addition or subtraction process (see Table 8). In our experiments, we primarily use the *random operands* counterfactual for baseline comparison, but we provide the additional counterfactuals for further analysis.

*Table 7.* Prompt templates for single-operator two-operand arithmetic operations.

| Template | Addition | Subtraction |
|---|---|---|
| 1 | Q: How much is $n_1$ plus $n_2$? A: | Q: How much is $n_1$ minus $n_2$? A: |
| 2 | Q: What is $n_1$ plus $n_2$? A: | Q: What is $n_1$ minus $n_2$? A: |
| 3 | Q: What is the result of $n_1$ plus $n_2$? A: | Q: What is the result of $n_1$ minus $n_2$? |
| 3 | The sum of $n_1$ and $n_2$ is: | The difference between $n_1$ and $n_2$ is: |
| 5 | $n_1 + n_2 =$ | $n_1 - n_2 =$ |
| 6 | $n_1 + n_2 =$ | $n_1 - n_2 =$ |

## D.3. Multiple-choice question answering (MCQA)

We expand the dataset of Wiegreffe et al. (2025), itself based on Norlund et al. (2021), by adding 102 additional instances from Paik et al. (2021) whose object group is "0", indicating that participants agreed on a prototypical color for that object—for example, that bananas are yellow. We randomly sample 3 incorrect colors from a set of 11 and pair them with the correct answer choice in a random position to create each instance. By design, each answer in this task is a single token

*Table 8.* An Arithmetic example and its 7 associated counterfactuals.

| Prompt / Counterfactual | Text | Correct Completion |
|---|---|---|
| Original Prompt | *The sum of 27 and 64 is:* | 91 |
| Random Operands | *The sum of 42 and 29 is:* | 71 |
| Different ones digit in operand 1 | *The sum of 24 and 64 is:* | 88 |
| Different ones digit in operand 2 | *The sum of 27 and 61 is:* | 88 |
| Different tens digit in operand 1 | *The sum of 47 and 64 is:* | 111 |
| Different tens digit in operand 2 | *The sum of 27 and 44 is:* | 71 |
| Different ones digit carry value | *The sum of 21 and 60 is:* | 81 |
| Different tens digit carry value | *The sum of 77 and 64 is:* | 141 |

(e.g., A, B, C, D).

In the paper, we report on 4-choice MCQA, as this is a standard number of choices and allows comparison with the ARC dataset. We also create versions of the dataset with 2, 3, 5, 6, 7, 8, 9, and 10 answer choices, in order to allow for future investigation into how mechanisms change as the result of having fewer or more choices.

**Counterfactuals.** We create two semantic counterfactuals and three format counterfactuals for each instance (and four combinations of these, resulting in 9 counterfactuals total). Semantic perturbations involve replacing the noun in the question (such as "banana") with a different noun from another instance in the same split, or the correct color of the noun mentioned in the question (such as "yellow") with another color (such as "brown"). The latter changes the correct answer; the former does not.

Format perturbations do not change the correct color itself, but do change the symbol that represents that color, and therefore, change the correct answer. We follow a similar design to Lieberum et al. (2023a). We change the position of the correct answer, the symbols representing the answer choices (i.e., 1/2/3/4 instead of A/B/C/D), or the letters (i.e., the randomly selected sequence E/Z/F/L instead of A/B/C/D). See Table 9 for an example dataset instance and its associated counterfactuals.

### D.4. AI2 Reasoning Challenge (ARC)

By design, each answer in this task is a single token (e.g., A, B, C, D), making the prompt format similar to MCQA (Appendix D.3).

**Counterfactuals.** The counterfactual types and generation process are identical to the MCQA counterfactual generation process (described in Appendix D.3). Due to the varying content of each ARC prompt and the lack of a token-level template between prompts, we include only the format-based counterfactuals, omitting *semantic* counterfactuals such as "Noun" and "Color" from MCQA. For an example prompt and its counterfactuals, see Table 10.

### D.5. Model Performance

For all tasks, we report accuracy given greedy generations in Table 1. For tasks that involve selecting between a fixed set of answer choices (IOI, MCQA, and ARC), we additionally report ranked-choice accuracy in Table 11. Ranked-choice scoring computes a model's prediction as the token that is assigned the highest probability within the set of answer choices; this is an upper bound on greedy generation performance. Ranked-choice scoring is more in line with the metric $m$ used for circuit localization (§3.1); greedy scoring is more in line with the prerequisite of causal variable localization (§4).

In Table 12 and Table 13 we report the results for each counterfactual type in the MCQA and ARC tasks, respectively. For the IOI and Arithmetic tasks, we found that due to the counterfactual format, all counterfactual types lead to the same performance as the original prompts (except for the "ABC" counterfactual in IOI, which has no correct answer).

*Table 9.* An MCQA example and its 9 associated counterfactuals.

| Prompt / Counterfactual | Text | Correct Completion |
|---|---|---|
| Original Prompt | *Question: Salmon meat is pink. What color is salmon meat?*
*A. gray\nB. black\nC. white\nD. pink\nAnswer:* | D |
| Noun | *Question: A banana is pink. What color is a banana?*
*A. gray\nB. black\nC. white\nD. pink\nAnswer:* | D |
| Color | *Question: Salmon meat is yellow. What color is salmon meat?*
*A. gray\nB. black\nC. white\nD. yellow\nAnswer:* | D |
| Noun+Color | *Question: A banana is yellow. What color is a banana?*
*A. gray\nB. black\nC. white\nD. yellow\nAnswer:* | D |
| Answer Position | *Question: Salmon meat is pink. What color is salmon meat?*
*A. gray\nB. black\nC. pink\nD. white\nAnswer:* | C |
| Symbol | *Question: Salmon meat is pink. What color is salmon meat?*
*1. gray\n2. black\n3. white\n4. pink\nAnswer:* | 4 |
| Random Letter | *Question: Salmon meat is pink. What color is salmon meat?*
*E. gray\nZ. black\nF. white\nL. pink\nAnswer:* | L |
| Answer Position + Random Letter | *Question: Salmon meat is pink. What color is salmon meat?*
*E. gray\nZ. black\nF. pink\nL. white\nAnswer:* | F |
| Answer Position + Symbol | *Question: Salmon meat is pink. What color is salmon meat?*
*1. gray\n2. black\n3. pink\n4. white\nAnswer:* | 3 |
| Answer Position + Color | *Question: Salmon meat is yellow. What color is salmon meat?*
*A. gray\nB. black\nC. yellow\nD. white\nAnswer:* | C |

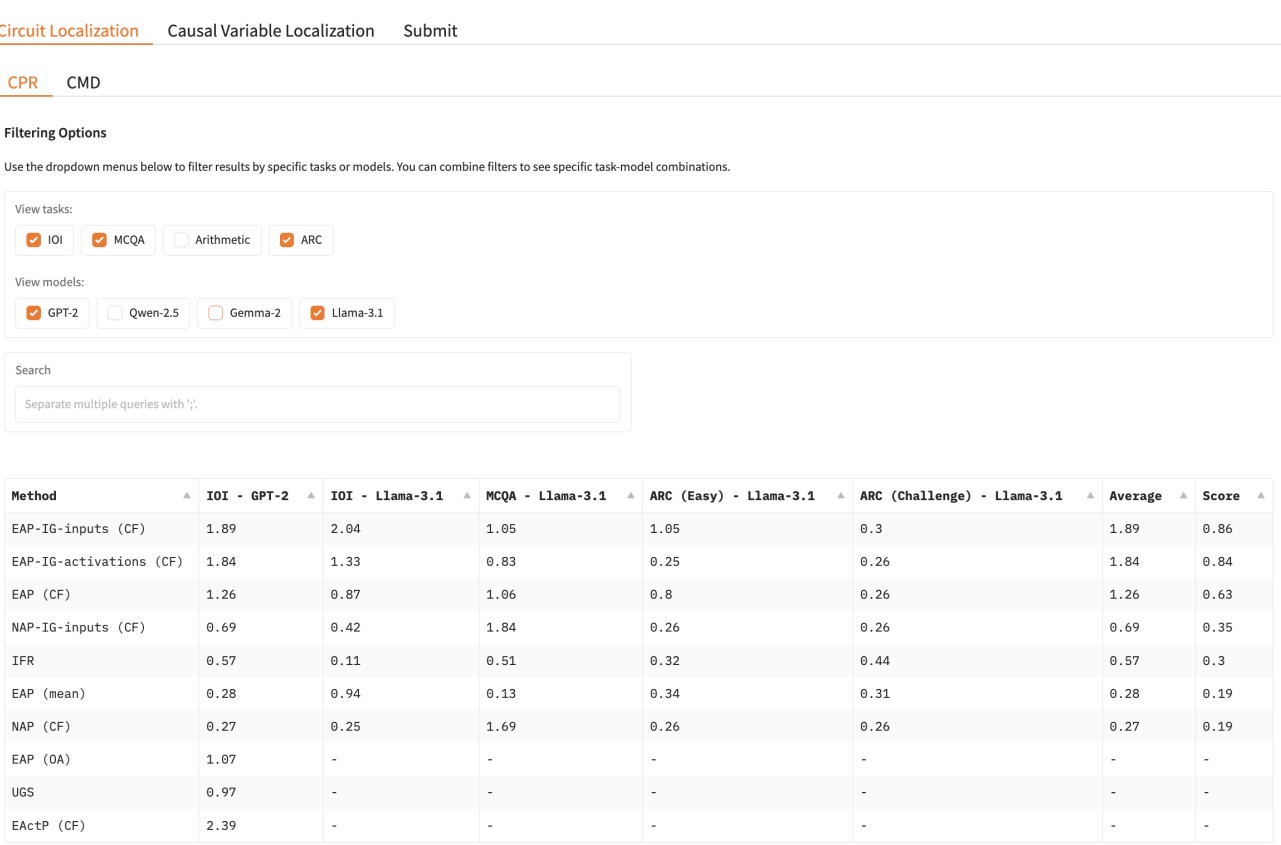

*Figure 4.* Leaderboard for the circuit localization track.

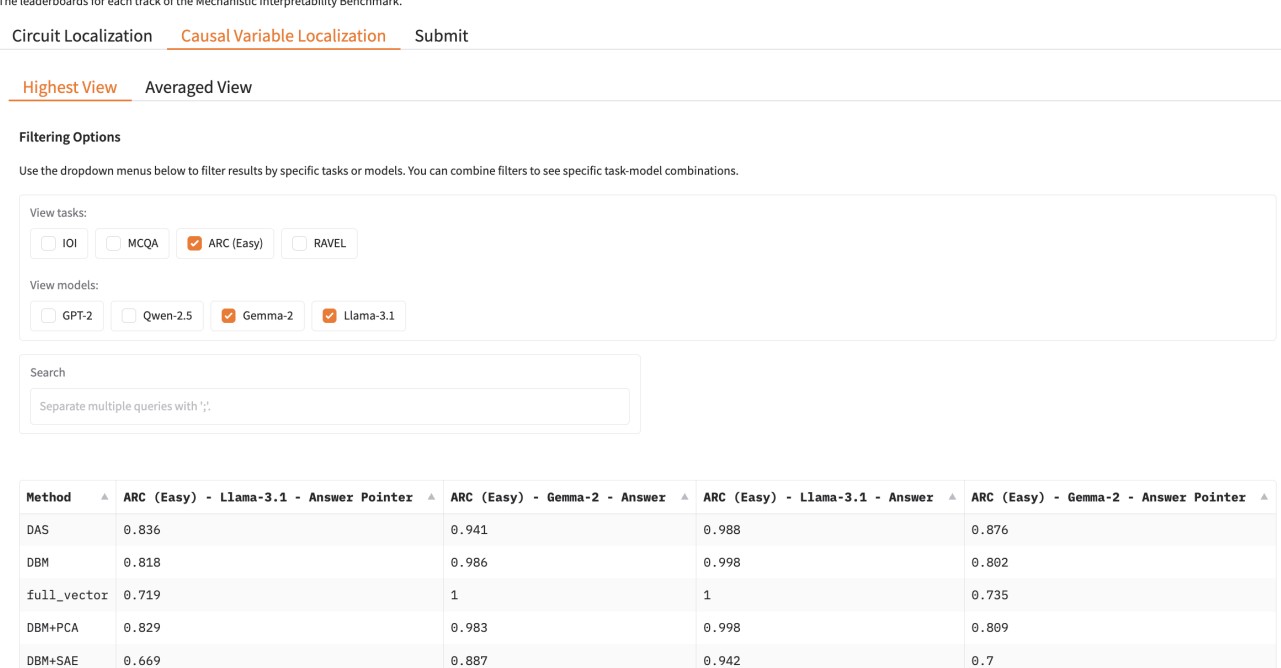

*Figure 5.* Leaderboard for the causal variable localization track.

*Table 10.* An ARC example and its 4 associated counterfactuals.

| Prompt / Counterfactual | Text | Correct Completion |
|---|---|---|
| Original Prompt | *Question: How does a tiger get stripes?* 
 *A. from its environment* 
 *B. from its food* 
 *C. from its offspring* 
 *D. from its parents* 
 *Answer:* | D |
| Answer Position | *Question: How does a tiger get stripes?* 
 *A. from its food* 
 *B. from its parents* 
 *C. from its environment* 
 *D. from its offspring* 
 *Answer:* | B |
| Symbol | *Question: How does a tiger get stripes?* 
 *1. from its environment* 
 *2. from its food* 
 *3. from its offspring* 
 *4. from its parents* 
 *Answer:* | 4 |
| Random Letter | *Question: How does a tiger get stripes?* 
 *D. from its environment* 
 *H. from its food* 
 *M. from its offspring* 
 *E. from its parents* 
 *Answer:* | E |
| Answer Position + Random Letter | *Question: How does a tiger get stripes?* 
 *D. from its food* 
 *H. from its parents* 
 *M. from its environment* 
 *E. from its offspring* 
 *Answer:* | H |

### D.6. Leaderboard

Here, we present screenshots of the MIB leaderboards. The leaderboards for both tracks are hosted on the same webpage; they are in separate tabs. The circuit localization track's leaderboard can be viewed in Figure 4. It has two tabs: one for CPR and one for CMD. Each row displays an Average score, which is a macroaverage of all scores in the row. Each row also displays a Score column, which is an average of the sigmoid of each score; we apply a sigmoid because each faithfulness value exists in a separate scale, where the lower and upper bounds may be different. Thus, to prevent any one column from dominating the score, we normalize each score to a [0, 1] range by applying a sigmoid to each CPR or CMD value before averaging. Users may filter rows based on model name and/or task name. After filtering, the Average and Score columns are recomputed dynamically. This allows users to compare performance at varying levels of granularity, or in cases where some methods are only tractable to run for a subset of the task/model combinations.

The causal variable localization track's leaderboard can be viewed in Figure 5. It displays results aggregated across all layers and token positions, *and* across counterfactual types. As in the circuit localization track, the Average is recomputed dynamically after filtering.

Our leaderboard will accept user submissions. To submit to the circuit localization track, a user must supply either

*Table 11.* Model performance for all models on the public test split of each analyzed task (0-shot) with ranked-choice accuracy.

| | IOI | MCQA | ARC (E) | (C) |
|---|---|---|---|---|
| Llama-3.1 8B | 1.00 | 0.92 | 0.93 | 0.79 |
| Gemma-2 2B | 1.00 | 1.00 | 0.79 | 0.60 |
| Qwen-2.5 0.5B | 1.00 | 1.00 | 0.73 | 0.58 |
| GPT-2 Small | 1.00 | 0.30 | 0.23 | 0.23 |

*Table 12.* Model accuracy (0-shot, greedy generation) on MCQA counterfactuals. N = Noun, C = Color, AP = Answer Position, S = Symbol, RL = Random Letter

| Model | N | C | N+C | AP | S | RL | AP+RL | AP + S | AP + C |
|---|---|---|---|---|---|---|---|---|---|
| Llama 3.1-8B | 0.70 | 0.72 | 0.96 | 0.94 | 0.94 | 0.90 | 0.98 | 0.98 | 0.72 |
| Gemma 2-2B | 0.92 | 1.00 | 1.00 | 1.00 | 0.98 | 0.70 | 0.74 | 1.00 | 0.98 |
| Qwen 2.5-0.5B | 1.00 | 1.00 | 1.00 | 1.00 | 1.00 | 0.52 | 0.60 | 1.00 | 0.98 |
| GPT2-Small | 0.00 | 0.06 | 0.00 | 0.02 | 0.00 | 0.00 | 0.00 | 0.00 | 0.02 |

(i) importance scores $\in \mathbb{R}$ on each node or edge, or (ii) 9 circuits of different sizes with membership in $\mathcal{C}$ given as Booleans. Recall from §3 that we use circuits containing varying percentages of edges; we will enforce this as a submission requirement for the circuit localization track. The smallest circuit can contain $k$ edges in the model, where $\frac{|\mathcal{C}_k|}{|\mathcal{N}|} \leq 0.001$. The second-smallest can contain any proportion of edges $k \leq 0.2$; the largest can contain $k \leq 0.5$, and so on.

For the causal variable localization track, a user must provide an invertible featurizer function $\mathcal{F}$ and token position functions specifying where in an input to apply them. These must be provided as Python scripts. The trained featurizer, inverse featurizer, and token indices must also be provided for each task/model combination. We will evaluate whether interchange interventions on the features and the variable result in the same behavior for each counterfactual dataset and average the results across them.

# E. Details on Circuit Localization Track

## E.1. Methods

**Mapping from scores to circuits.** All of the techniques that we benchmark produce a set of scores, which must then be mapped to circuits. To find a circuit with $n$ edges, we can simply take the top-$n$ edges by score—what we call the *top-$n$* method for constructing circuits. However, this approach can often result in a circuit without an end-to-end pathway from inputs to outputs. Alternatively, we can perform a *greedy search* starting from the logits as follows. Let our circuit $C = (V_C, V_E) = (\{logits\}, \emptyset)$. Then for $i = 1, \ldots, n$ add the highest-$\hat{IE}$ edge connected to $V_C$ that is not currently in $V_E$, to $V_E$; add its parent to $V_C$. For simplicity, we use top-$n$, except in cases where it tends to not work well; in practice, greedy circuit construction is only needed for good performance when using information flow routes.

When deciding which components to use, we can either add to the circuit the edges with the highest score. Or, we can first take the absolute value of each score, adding the highest-*magnitude* scores. Adding the highest-scoring components is more likely to yield components that perform the task well, and is better suited to the CPR metric. Adding the highest-magnitude components is more likely to yield components that have *any* strong effect on task performance, and is better suited to the CMD metric. Thus, when we report scores, we use high-value scoring to construct circuits for CPR, and high-magnitude scoring to construct circuits for CMD.

## E.2. Baselines

**Attribution patching.** Edge attribution patching is computed as follows. Let $(u, v)$ be an edge from component $u$ to $v$, $a_u$ and $a'_u$ be the output activations of $u$ on normal and counterfactual inputs, $a_v$ be the input activations of $v$, and $m$ be our

*Table 13.* Model accuracy (0-shot, greedy generation) on ARC counterfactuals. AP = Answer Position, S = Symbol, RL = Random Letter.

| Easy/Challenge | Model | AP | S | RL | AP+RL |
|---|---|---|---|---|---|
| Easy | Llama 3.1-8B | 0.93 | 0.91 | 0.86 | 0.85 |
| | Gemma 2-2B | 0.78 | 0.78 | 0.57 | 0.55 |
| | Qwen 2.5-0.5B | 0.73 | 0.66 | 0.26 | 0.22 |
| | GPT2-Small | 0.03 | 0.00 | 0.00 | 0.01 |
| Challenge | Llama 3.1-8B | 0.79 | 0.77 | 0.69 | 0.67 |
| | Gemma 2-2B | 0.60 | 0.62 | 0.43 | 0.41 |
| | Qwen 2.5-0.5B | 0.56 | 0.51 | 0.18 | 0.19 |
| | GPT2-Small | 0.04 | 0.00 | 0.01 | 0.00 |

model performance metric. EAP estimates the indirect effect as

$$\hat{\text{IE}} = (a_u - a_u')\frac{\partial m}{\partial a_v}\Big|_x. \tag{3}$$

That is, we multiply the change in activation of $u$ by the gradient (slope) of the metric $m$ with respect to $v$'s input, on normal inputs $x$.

When performing NAP (at the node level rather than edge level), the $\hat{\text{IE}}$ of a node can be computed as in Eq. 3, but replacing $\frac{\partial m}{\partial a_v}$ with $\frac{\partial m}{\partial a_u}$. Note that we always run NAP at the *neuron* granularity, and not the submodule granularity; this is because at smaller circuit sizes, including just one submodule puts us over the size threshold, meaning we have multiple points at which the circuit is empty.

**Attribution patching with integrated gradients.** Edge attribution patching with IG (EAP-IG) is defined as follows:

$$\hat{\text{IE}} = (a - a') \cdot \frac{1}{Z}\sum_{z=0}^{Z}\frac{\partial m(a' + \frac{z}{Z}\cdot(a - a'))}{\partial a}\Big|_x. \tag{4}$$

That is, given input $x$, we compute $\frac{\partial d}{\partial a}$ at $Z$ intermediate points between $a$ and $a'$. At each intermediate point, we intervene on $a$, replacing its activation with what it would have been at the intermediate point. Using this new activation, we recompute $m$, and backpropagate from that to obtain a new gradient value. We take the mean over these gradient values to obtain a more accurate estimate of the slope of $m$ w.r.t. $a$. This slope is then multiplied by the change in $a$ as before.

EAP-IG-inputs operates under a similar intuition. The key difference is that, instead of interpolating between intermediate activations at the target neuron, we only interpolate between intermediate activations at the input embeddings, and allow the network to compute the activations for the target component naturally given each intermediate input embedding. EAP-IG-activations therefore requires us to perform this interpolation for each layer separately; EAP-IG-inputs only requires us to perform this interpolation once at the inputs.

**IFR.** We adapt IFR to output importance scores for our computational graph as follows. Let $a_v$ be the *input* to $v$; if $\mathcal{U}$ is the set of nodes with edges to $v$, and $a_u$ is the *output* of a node $u \in U$, then the importance of a given $u$ to $v$ is

$$\text{imp}(u, v) = \frac{\max(||a_v||_1 - ||a_u - a_v||_1, 0)}{\sum_{u' \in \mathcal{U}}\max(||a_v||_1 - ||a_{u'} - a_v||_1, 0)}. \tag{5}$$

Because important scores are normalized (for any given node, the sum of the scores of edges to it will be 1), we cannot apply a top-$n$ procedure to find IFR circuits; we must use greedy search.

**Uniform Gradient Sampling.** UGS maintains a parameter $\tilde{\theta}_{(u,v)}$ for each edge $(u, v)$, where $\theta_{(u,v)} = (1 + \exp(-\tilde{\theta}_{(u,v)}))^{-1}$ represents the estimated probability of $(u, v)$ being part of the circuit determined by the pruning mask. The sampling frequency for $\alpha_{(u,v)}$ is determined by $w(\theta_{(u,v)}) = \theta_{(u,v)}(1 - \theta_{(u,v)})$. Specifically, $\alpha_{(u,v)} \sim \text{Unif}(0, 1)$ with probability $w(\theta_{(u,v)})$, $\alpha_{(u,v)} = 1$ with probability $\theta_{(u,v)} - \frac{1}{2}w(\theta_{(u,v)})$, and $\alpha_{(u,v)} = 0$ with probability $1 - \theta_{(u,v)} - \frac{1}{2}w(\theta_{(u,v)})$. We use $\theta_{(u,v)}$ as the importance scores when constructing circuits.

*Table 14.* CPR scores across circuit localization methods and ablation types. All evaluations were performed using counterfactual ablations. Higher scores are better. Arithmetic scores are averaged across addition and subtraction; see Table 17 for separate scores. We **bold** and underline the best and second-best methods per column, respectively.

| | IOI | | | | Arithmetic | MCQA | | | ARC (E) | | ARC (C) |
|---|---|---|---|---|---|---|---|---|---|---|---|
| Method | GPT-2 | Qwen-2.5 | Gemma-2 | Llama-3.1 | Llama-3.1 | Qwen-2.5 | Gemma-2 | Llama-3.1 | Gemma-2 | Llama-3.1 | Llama-3.1 |
| Random | 0.25 | 0.28 | 0.30 | 0.25 | 0.25 | 0.27 | 0.32 | 0.26 | 0.32 | 0.26 | 0.25 |
| EActP (CF) | **2.30** | 1.21 | - | - | - | 0.85 | - | - | - | - | - |
| EAP (mean) | 0.29 | 0.71 | 0.68 | 0.98 | 0.35 | 0.29 | 0.33 | 0.13 | 0.26 | 0.34 | 0.80 |
| EAP (CF) | 1.20 | 0.26 | 1.29 | 0.85 | 0.55 | 0.85 | 1.49 | 1.00 | 1.08 | 0.80 | 0.82 |
| EAP (OA) | 0.95 | 0.70 | - | - | - | 0.29 | - | - | - | - | - |
| EAP-IG-inputs (CF) | 1.85 | **1.63** | **3.20** | **2.08** | **0.99** | 1.16 | 1.64 | 1.05 | 1.53 | **1.04** | **0.98** |
| EAP-IG-activations (CF) | 1.82 | 1.63 | 2.07 | 1.60 | 0.98 | 0.77 | 1.57 | 0.79 | **1.70** | 0.71 | 0.63 |
| NAP (CF) | 0.28 | 0.30 | 0.30 | 0.26 | 0.27 | 0.38 | 1.47 | 1.69 | 1.01 | 0.26 | 0.26 |
| NAP-IG (CF) | 0.76 | 0.29 | 1.52 | 0.42 | 0.39 | 0.77 | **1.71** | **1.87** | 1.53 | 0.26 | 0.26 |
| IFR | 0.58 | 0.31 | 0.25 | 0.09 | 0.89 | 0.40 | 0.38 | 0.52 | 0.34 | 0.36 | 0.24 |
| UGS | 0.97 | 0.98 | - | - | - | **1.17** | - | - | - | - | - |

The loss function comprises two components: (1) a performance metric that measures the discrepancy between the original model's predictions and the output of the partially ablated model (here, KL divergence); and (2) a regularization term that controls the sparsity of the subgraph determined by the pruning mask. The balance between these components is governed by a hyperparameter $\lambda$. For our experiments, we set $\lambda = 10^{-3}$, chosen through a hyperparameter search over $\{10^{-2}, 10^{-3}, \ldots, 10^{-7}\}$ using a validation set. All other hyperparameters were left at their default values, as specified in Li & Janson (2024).

**Optimal ablations.** In optimal ablations (Li & Janson, 2024), rather than taking an activation from an example-dependent counterfactual input, we learn an ablation vector $\mathbf{a}$ that is not dependent on the original input. Given submodule $u$ taking activations $\mathbf{u}$, we initialize the ablation vector to the mean of $\mathbf{u}$ over the task dataset $\mathcal{D}$. Then, we optimize $\mathbf{a}$ via gradient descent to minimize

$$\arg \min_{\mathbf{a}} \mathcal{L}(\mathcal{N}, \mathcal{D}, \mathrm{do}(\mathbf{u} = \mathbf{a})), \tag{6}$$

where $\mathcal{L}$ is the cross-entropy (language modeling) loss on the task dataset $\mathcal{D}$ when we set $\mathbf{u}$ to $\mathbf{a}$. We pre-compute this vector for all $u$ in $\mathcal{N}$, and then use these vectors as the counterfactual activations during circuit discovery.

We use initial learning rate $1 \times 10^{-3}$ and batch size 20. We train for up to 1000 steps on the train split of the task dataset. We compute loss on the validation split every 50 steps; if the validation loss does not improve from its best value after 150 steps, we stop early and save the ablation vector from the best evaluation step.

### E.3. Further Circuit Localization Results

Table 14 presents CPR scores for all valid task-model combinations. Trends are largely similar to those from the CMD table, except that EAP and UGS are less competitive with EAP-IG-inputs.

We provide scores for all methods where possible. UGS has significant memory requirements; running it on an 80G GPU is not possible for larger models, even when reducing the batch size to 1. Edge activation patching (EActP) and optimal ablations–based methods do not scale well time-wise with model size; the number of edges multiplies significantly, meaning that we must iterate over many more components. We do not include methods that take over 1 week to run.[12]

In Table 17 we provide scores for each arithmetic operator separately.

## F. Details on InterpBench Model Training

Faithfulness is a fuzzy and unbound metric. Ideally, we would like to know which edges or nodes are in the circuit in advance, such that we can compute more precise metrics such as precision and recall. Inspired by InterpBench (Gupta et al., 2024), we train a transformer model closely following their methods. The model we use was explicitly trained to predict the

---

[12]This is an arbitrary threshold. We do not restrict users from submitting methods that take this long to run if they so choose.

*Table 15.* CMD scores for the *private* test set across circuit localization methods and ablation types (lower is better). All evaluations were performed using counterfactual ablations. Arithmetic scores are averaged across addition and subtraction. We **bold** and underline the best and second-best methods per column, respectively.

| Method | IOI | | | | Arithmetic | MCQA | | | ARC (E) | | ARC (C) |
|---|---|---|---|---|---|---|---|---|---|---|---|
| | GPT-2 | Qwen-2.5 | Gemma-2 | Llama-3.1 | Llama-3.1 | Qwen-2.5 | Gemma-2 | Llama-3.1 | Gemma-2 | Llama-3.1 | Llama-3.1 |
| Random | 0.75 | 0.72 | 0.70 | 0.75 | 0.75 | 0.73 | 0.68 | 0.74 | 0.68 | 0.73 | 0.75 |
| EActP (CF) | **0.01** | 0.48 | - | - | - | 0.35 | - | - | - | - | - |
| EAP (mean) | 0.27 | 0.24 | 0.28 | 0.04 | 0.07 | 0.21 | 0.19 | 0.17 | 0.22 | 0.19 | 0.20 |
| EAP (CF) | 0.03 | 0.15 | 0.06 | **0.01** | 0.01 | 0.12 | 0.08 | **0.12** | **0.04** | 0.20 | **0.19** |
| EAP (OA) | 0.31 | 0.17 | - | - | - | 0.11 | - | - | - | - | - |
| EAP-IG-inp. (CF) | 0.03 | 0.02 | 0.04 | **0.01** | 0.01 | 0.08 | **0.06** | 0.14 | **0.04** | **0.10** | 0.22 |
| EAP-IG-act. (CF) | 0.02 | **0.01** | **0.03** | **0.01** | **0.00** | **0.05** | 0.07 | **0.12** | **0.04** | 0.30 | 0.38 |
| NAP (CF) | 0.36 | 0.33 | 0.40 | 0.30 | 0.28 | 0.24 | 0.29 | 0.36 | 0.33 | 0.69 | 0.69 |
| NAP-IG (CF) | 0.25 | 0.19 | 0.29 | 0.18 | 0.17 | 0.18 | 0.29 | 0.33 | 0.27 | 0.67 | 0.67 |
| IFR | 0.43 | 0.70 | 0.75 | 0.89 | 0.22 | 0.60 | 0.62 | 0.49 | 0.66 | 0.63 | 0.53 |
| UGS | 0.04 | 0.02 | - | - | - | - | 0.19 | - | - | - | - |

*Table 16.* CPR scores for the *private* test set across circuit localization methods and ablation types. All evaluations were performed using counterfactual ablations. Higher scores are better. Arithmetic scores are averaged across addition and subtraction. We **bold** and underline the best and second-best methods per column, respectively.

| Method | IOI | | | | Arithmetic | MCQA | | | ARC (E) | | ARC (C) |
|---|---|---|---|---|---|---|---|---|---|---|---|
| | GPT-2 | Qwen-2.5 | Gemma-2 | Llama-3.1 | Llama-3.1 | Qwen-2.5 | Gemma-2 | Llama-3.1 | Gemma-2 | Llama-3.1 | Llama-3.1 |
| Random | 0.25 | 0.28 | 0.30 | 0.25 | 0.25 | 0.27 | 0.32 | 0.26 | 0.32 | 0.26 | 0.25 |
| EActP (CF) | **2.39** | 1.20 | - | - | - | 0.87 | - | - | - | - | - |
| EAP (mean) | 0.28 | 0.34 | 0.64 | 0.94 | 0.34 | 0.29 | 0.34 | 0.13 | 0.26 | 0.34 | 0.31 |
| EAP (CF) | 1.26 | 0.27 | 1.31 | 0.87 | 0.54 | 0.84 | 1.48 | 1.06 | 1.08 | 0.80 | 0.26 |
| EAP (OA) | 1.07 | 0.75 | - | - | - | 0.29 | - | - | - | - | - |
| EAP-IG-inputs (CF) | 1.89 | **1.73** | **3.03** | **2.04** | **0.98** | **1.61** | 1.05 | 0.95 | 1.53 | **1.05** | 0.31 |
| EAP-IG-activations (CF) | 1.84 | 1.61 | 2.34 | 1.33 | **0.98** | 0.76 | 1.53 | 0.83 | **1.71** | 0.27 | 0.28 |
| NAP (CF) | 0.27 | 0.30 | 0.29 | 0.25 | 0.26 | 0.38 | 1.46 | 1.69 | 1.02 | 0.26 | 0.26 |
| NAP-IG (CF) | 0.69 | 0.29 | 1.42 | 0.42 | 0.38 | 0.77 | **1.68** | **1.84** | 1.54 | 0.26 | 0.26 |
| IFR | 0.57 | 0.30 | 0.25 | 0.11 | 0.89 | 0.40 | 0.38 | 0.51 | 0.34 | 0.37 | **0.47** |
| UGS | 0.97 | 1.00 | - | - | - | 1.17 | - | - | - | - | - |

indirect objects in the IOI dataset (App. D.1), and implements a simplified version of the IOI circuit described by Gupta et al. (2024).

The model has 6 layers and 4 heads per layer, $d_{model} = 64$, and $d_{head} = 16$. It was trained with mini-batches of varying lengths using left padding. We performed hyperparameter sweeps to find the best weights for the SIIT algorithm. We use the three-losses variant of SIIT; see Gupta et al. (2024) for details. The final model was trained for 70 hours on a single H100 GPU.

*Table 17.* CPR and CMD scores for Llama-3.1 on the arithmetic public test sets, separated by operator. Scores are generally similar across operators, and methods follow similar rankings regardless of which operators are used. A notable exception is EAP-IG-activations, where CPR scores are significantly different.

| Method | Arithmetic $(+)$ | | Arithmetic $(-)$ | |
|---|---|---|---|---|
| | CPR $(\uparrow)$ | CMD $(\downarrow)$ | CPR $(\uparrow)$ | CMD $(\downarrow)$ |
| Random | 0.25 | 0.25 | 0.75 | 0.75 |
| EAP (mean) | 0.40 | 0.07 | 0.31 | 0.07 |
| EAP (CF) | 0.49 | 0.01 | 0.61 | 0.01 |
| EAP-IG-inputs | 0.96 | 0.00 | 1.03 | 0.00 |
| EAP-IG-activations | 0.98 | 0.00 | 0.99 | 0.00 |
| NAP (CF) | 0.26 | 0.29 | 0.27 | 0.27 |
| NAP-IG (CF) | 0.43 | 0.18 | 0.34 | 0.18 |
| IFR | 0.90 | 0.24 | 0.87 | 0.20 |

# G. Details on Causal Variable Localization Track

## G.1. Causal Abstraction Analysis

**Causal Models and Interventions**  A deterministic causal model $\mathcal{H}$ has *variables* that take on *values*. Each variable has a *mechanism* that determines the value of the variable based on the values of *parent variables*. Variables without parents, denoted $\mathbf{X}$, can be thought of as inputs that determine the setting of all other variables, denoted $\mathcal{H}(\mathbf{x})$. A *hard intervention* $X \leftarrow x$ overrides the mechanisms of variable $X$, fixing it to a constant value $x$.

**Interchange Interventions**  We perform *interchange interventions* (Vig et al., 2020; Geiger et al., 2020) where a variable (or set of features) $X$ is fixed to be the value it would take on if the LM were processing *counterfactual input c*. We write $X \leftarrow \mathsf{Get}(\mathcal{H}(c), X)$ where $\mathsf{Get}(\mathcal{H}(c), X)$ is the value of variable $X$ when $\mathcal{H}$ processes input $c$. In experiments, we will feed a *base input* $b$ to a model under an interchange intervention $\mathcal{H}_{X \leftarrow \mathsf{Get}(\mathcal{H}(c), X))}(b)$.

**Featurizing Hidden Vectors**  The dimensions of hidden vectors are not an ideal unit of analysis (Smolensky, 1986), and so it is typical to *featurize* a hidden vector using some invertible function, e.g., an orthogonal matrix, to project a hidden vector into a new variable space with more interpretable dimensions called "features"(Geiger et al., 2024a; Huang et al., 2024a). A feature intervention $\Pi \leftarrow \Pi$ edits the mechanism of a hidden vector $\mathbf{h}$ to fix the value of features $\Pi$ to $\Pi$.

**Alignment**  The LM is a *low-level causal model* $\mathcal{N}$ where variables are dimensions of hidden vectors and the hypothesis about LM structure is a *high-level causal model* $\mathcal{H}$. An *alignment* assigns each high-level variable $X$ to features of a hidden vector $\Pi_{\mathbf{h}}^X$, e.g., orthogonal directions in the activation space of $\mathbf{h}$. To evaluate an alignment, we perform intervention experiments to evaluate whether high-level interventions on the variables in $\mathcal{H}$ have the same effect as interventions on the aligned features in $\mathcal{N}$.

**Causal Abstraction**  We use interchange interventions to reveal whether the hypothesized causal model $\mathcal{H}$ is an abstraction of an LM $\mathcal{N}$. To simplify, assume both models share an input and output space. The high-level model $\mathcal{H}$ is an abstraction of the low-level model $\mathcal{N}$ under a given alignment when each high-level interchange intervention and the aligned low-level intervention result in the same output. For a high-level intervention on $X$ aligned with low-level features $\Pi_{\mathbf{h}}^X$ with a counterfactual input $c$ and base input $b$, we write

$$\mathsf{GetOutput}(\mathcal{N}_{\Pi_{\mathbf{h}}^X \leftarrow \mathsf{Get}(\mathcal{N}(c), \Pi_{\mathbf{h}}^X))}(b)) = \mathsf{GetOutput}(\mathcal{H}_{X \leftarrow \mathsf{Get}(\mathcal{H}(c), X))}(b)) \qquad (7)$$

If the low-level interchange intervention on the LM produces the same output as the aligned high-level intervention on the algorithm, this is a piece of evidence in favor of the hypothesis. This extends naturally to multi-variable interventions (Geiger et al., 2024a).

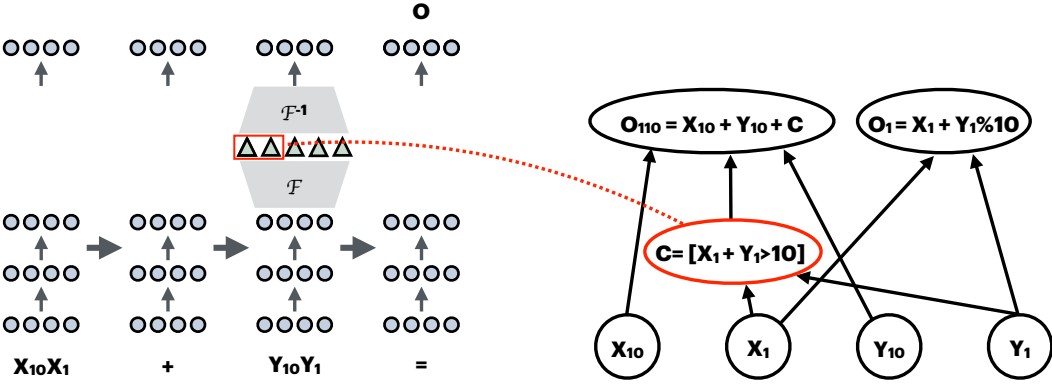

(a) **Arithmetic Task Submission.** Users submit an alignment between the carry-the-one variable $X_{\text{Carry}}$ in a high-level causal model $\mathcal{H}_+$ and two features $\Pi_{X_{\text{Carry}}}$ of the neural network's residual stream at the second number token.

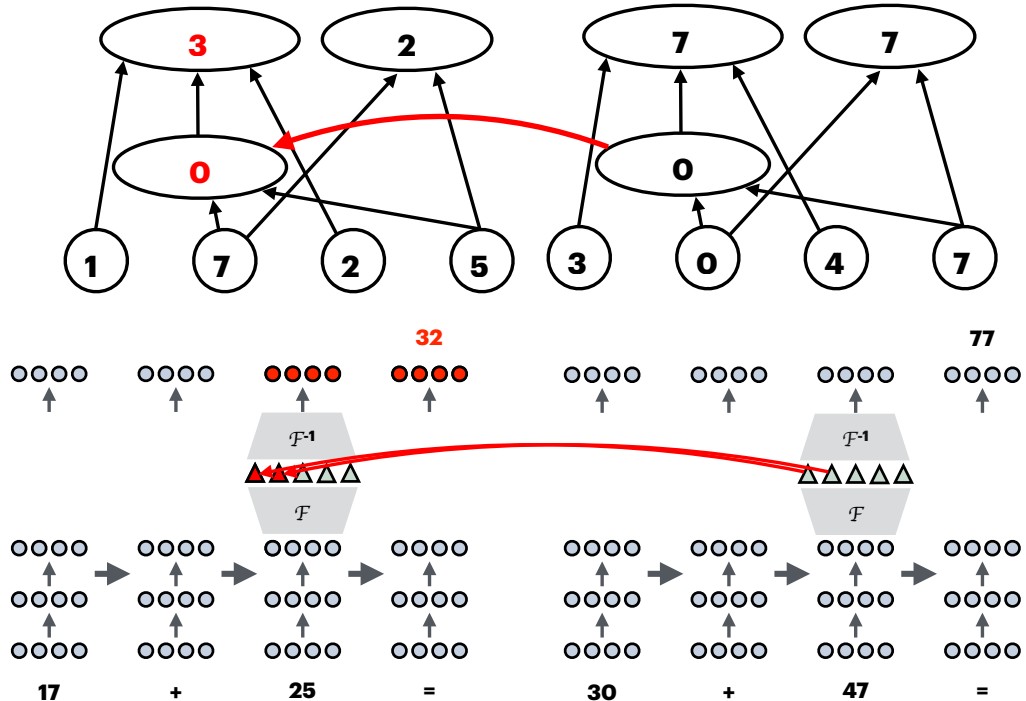

(b) **Arithmetic Task Evaluation.** An aligned interchange intervention with base input "17+25=" and counterfactual input "30+47=". At the high-level, the interchange intervention $\mathcal{H}_{X_{\text{Carry}} \leftarrow \text{Get}(\mathcal{H}(30+47), X_{\text{Carry}})}(17 + 25)$ fixes the carry-the-one variable $X_{\text{Carry}}$ to the value 0 (from the counterfactual) instead of its natural value 1 (from the base input), causing the causal model to output "32" instead of "42". At the low-level, the interchange intervention $\mathcal{N}_{\Pi_{X_{\text{Carry}}} \leftarrow \text{Get}(\mathcal{N}(30+47), \Pi_{X_{\text{Carry}}})}(17 + 25)$ fixes the aligned features $\Pi_{X_{\text{Carry}}}$ of the LM to the value they take on when the LM is run on the counterfactual input. The low-level output after intervention "32" is equal to the high-level output after intervention, which is a piece of evidence supporting the hypothesized alignment between the carry-the-one variable and the identified neural network features. The faithfulness metrics aggregate these individual experiments across base-counterfactual input pairs.

*Figure 6.* A schematic of the causal variable localization track submission and evaluation. Users submit an alignment between a high-level causal variable $X$ and hidden vector features $\Pi_X$ in an LM (top). In evaluations, aligned interchange interventions are performed with base and counterfactual inputs $(b, c)$ on the high-level causal model $\mathcal{H}_{X \leftarrow \text{Get}(\mathcal{H}(c), X)}(b)$ and the low-level neural network $\mathcal{N}_{\Pi_X \leftarrow \text{Get}(\mathcal{N}(c), \Pi_X)}(b)$. The more similar LM output under intervention is to the causal model output under intervention, the more faithfully the causal model abstracts the LM (bottom).

**Graded Faithfulness Metric**    We construct *counterfactual datasets* for each causal variable where an example consists of a base prompt and a counterfactual prompt . The *counterfactual label* is the expected output of the algorithm after the high-level interchange intervention, i.e., the right-side of Equation 7. The interchange intervention accuracy is the proportion of examples for which Equation 7 holds, i.e., the degree to which $\mathcal{H}$ faithfully abstracts $\mathcal{N}$.

## G.2. Aligning Unsupervised Features to Causal Variables

In our experiments, we use a variety of unsupervised methods for featurizing hidden vectors in LMs, including principal component analysis (PCA), sparse autoencoders (SAE), and simply taking standard dimensions of the hidden vector as features. For a variable $X$ in the high-level causal model $\mathcal{H}$, we learn a set of features $\Pi_{\mathbf{h}}^X$ of a hidden vector $\mathbf{h}$ of the LM $\mathcal{N}$ using Differential Binary Masking (DBM) (Cao et al., 2020; 2022; Csordás et al., 2021; Davies et al., 2023). Given base input $b$ and counterfactual input $c$, we train a mask $\mathbf{m} \in [0, 1]^{|\Pi_{\mathbf{h}}|}$ on the objective

$$\mathsf{CE}\Big( \mathsf{GetLogits}\big( \mathcal{N}_{\Pi_{\mathbf{h}} \leftarrow \mathbf{m} \circ \mathsf{Get}(\mathcal{N}(c), \Pi_{\mathbf{h}}))}(b) \big), \mathsf{GetLogits}\big( \mathcal{H}_{X \leftarrow \mathsf{Get}(\mathcal{H}(c), X))}(b) \big) \Big) \tag{8}$$

**Principal Component Analysis.**    Principal Component Analysis (PCA) serves as an unsupervised dimensionality reduction technique (Tigges et al., 2023; Marks & Tegmark, 2024). For a vector set $(\mathcal{V} \subset \mathbb{R}^n)$ where $(|\mathcal{V}| > n)$, PCA determines orthogonal unit vectors $\begin{bmatrix} \mathbf{p}_1 & \ldots & \mathbf{p}_n \end{bmatrix}$. We employ the principal components' orthogonal matrix as featurizer $\mathcal{F}$, mapping neurons into a more interpretable lower-dimensional space. Given PCA's unsupervised nature, which doesn't inherently specify component information, we use differential binary masking to select principal components that best abstracted by a causal variable.

**Sparse Autoencoders.**    Sparse Autoencoders (SAE) employ an autoencoder architecture to transform neural activations into a sparse, higher-dimensional feature space before reconstruction (Bricken et al., 2023; Huben et al., 2024). Our implementation utilizes the GemmaScope (Lieberum et al., 2024) and LlamaScope (He et al., 2024) SAE collections. A key consideration is that featurizer invertibility requires inclusion of the SAE reconstruction loss. Consequently, all SAE feature interventions incorporate the base input's reconstruction error. As with PCA, sparse autoencoders produce unsupervised features without inherent interpretability. We address this by implementing the previously described differential binary masking approach on SAE features (Chaudhary & Geiger, 2024).

## G.3. Distributed Alignment Search

**Distributed Alignment Search.**    Distributed Alignment Search (DAS) (Geiger et al., 2024b) operates as a supervised featurization technique that identifies a linear subspace within the model's representation space. The method utilizes an orthogonal matrix $Q$ of size $n \times n$, written as $Q = [\mathbf{u}_1 \ldots \mathbf{u}_n]$. This transformation matrix converts the original representation into a new coordinate system through $\mathcal{F}(\mathbf{h}) = Q^\top \mathbf{h}$. The feature subset $\Pi_{\mathbf{h}}$ is extracted from the first $k$ dimensions of this transformed space, where $k$ serves as an adjustable hyperparameter. The optimization of matrix $Q$ minimizes the following loss:

$$\mathcal{L} = \mathsf{CE}(\mathcal{H}_{X \leftarrow \mathsf{Get}(\mathcal{H}(c), X)}(b), \mathcal{N}_{\Pi_{\mathbf{h}}^X \leftarrow \mathsf{Get}(\mathcal{N}(c), \Pi_{\mathbf{h}}^X)}(b))$$

To manage computational efficiency, rather than computing the complete matrix $Q$, we learn only the $k$ orthogonal vectors that constitute feature $\Pi_{\mathbf{h}}^X$. Our implementation utilizes the pyvene library (Wu et al., 2024), training the featurizer on base-counterfactual pairs with interchange interventions.

### G.4. Hyperparameters

**Learning rate and regularization**    The learning rate used across models and tasks was 0.01, except for IOI which we used learning rate of 1.0. No regularization loss terms were used.

**Epochs and batch size.**    For RAVEL, we train for one epoch of ≈30k examples with a batch size of 128 for Llama and 32 for Gemma. For MCQA, we train for 8 epochs on ≈300 examples with a batch size of 64. For ARC (easy), we train for 2 epochs of ≈9k examples and a batch size of 48 with Gemma and for 1 epoch with a batch size of 16 for Llama. For the two-digit addition task, we train for 1 epoch on ≈30k examples with a batch size of 256. For IOI, we train for one epoch on ≈30k examples.

**DAS dimensionality.**    The dimensionality of DAS was set at 16 for the ordering ID $X_{\text{Order}}$ and carry-the-one variable $X_{\text{Carry}}$. The DAS dimensionality for $S_{\text{Tok}}$ and $S_{\text{Pos}}$ are 32. The $O_{\text{Answer}}$ variable in MCQA and ARC (Easy) has a DAS dimensionality of half the residual stream for their respective model, because token embeddings live in a higher dimensional space. The RAVEL task which had the dimensionality of an eighth of the residual stream, according to the experiments from (Huang et al., 2024a).

**Masking parameters.**    For the masking methods, the temperature schedule used begins at 1.0 and approaches 0.01.

### G.5. High-level Causal Models and Experimental Details for Each Task

#### G.5.1. MULTIPLE-CHOICE QUESTION ANSWERING

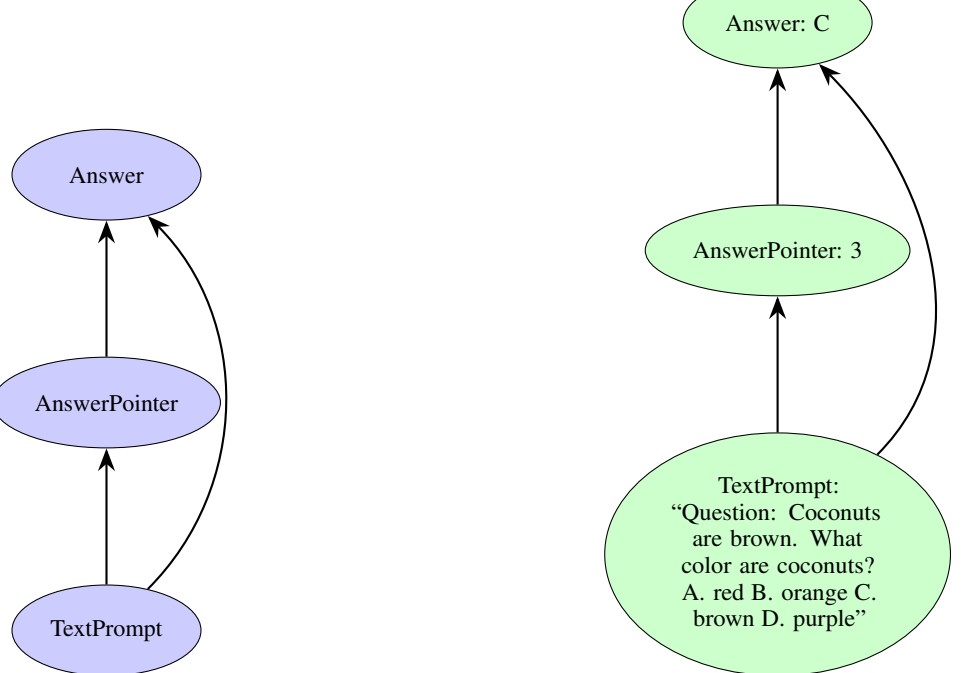

(a) General causal model for multiple choice question answering        (b) Specific example of the causal process

*Figure 7.* Causal model for multiple choice question answering. The model operates through a two-step mechanism: first, the TextPrompt is processed to generate an AnswerPointer that identifies the position of the correct answer in the options list. Second, this AnswerPointer is used to extract the corresponding letter label (A, B, C, or D) as the final Answer. This mechanism separates the reasoning process (identifying which option is correct) from the answer extraction (converting position to label).

We define the causal model for multiple-choice question answering, including one for the ARC dataset, in Figure 7. It comprises two variables, as illustrated in Figure 7a: 1) $X_{\text{Order}}$: It takes the text prompt as input and outputs a pointer that encodes the position of the correct label. 2) $O_{\text{Answer}}$: It receives the answer pointer as input and dereferences it by retrieving

the corresponding token value from the text prompt. For instance, consider the text prompt shown in Figure 7b. First, the $X_{\text{Order}}$ variable identifies the position of the correct option—position 3. Then, the $O_{\text{Answer}}$ variable uses this information to locate and extract the value of the third option from the prompt—i.e., $C$—which becomes the final output. Similar high-level causal models have also been proposed in prior work (Lieberum et al., 2023b; Prakash et al., 2024).

We conduct two interchange intervention experiments—one for each variable in the causal model—to align the LM's internal representations with those of the causal model. To align the $X_{\text{Order}}$ variable, we create counterfactual examples where the position of the correct option is altered, while the option label remains the same alphabetically. Conversely, to align the $O_{\text{Answer}}$ variable, we generate counterfactuals in which the position of the correct option is fixed, but the option label is replaced with a different letter.

Figures 8 and 9 show the alignment results of the $X_{\text{Order}}$ variable in the Gemma model using full vector patching and the DAS method, respectively. Both results demonstrate that the $X_{\text{Order}}$ information shifts from the correct symbol token position to the last token position in the middle layers. This behavior aligns with our hypothesized causal model, in which the model first identifies the position of the correct option, before dereferencing it to fetch token value information.

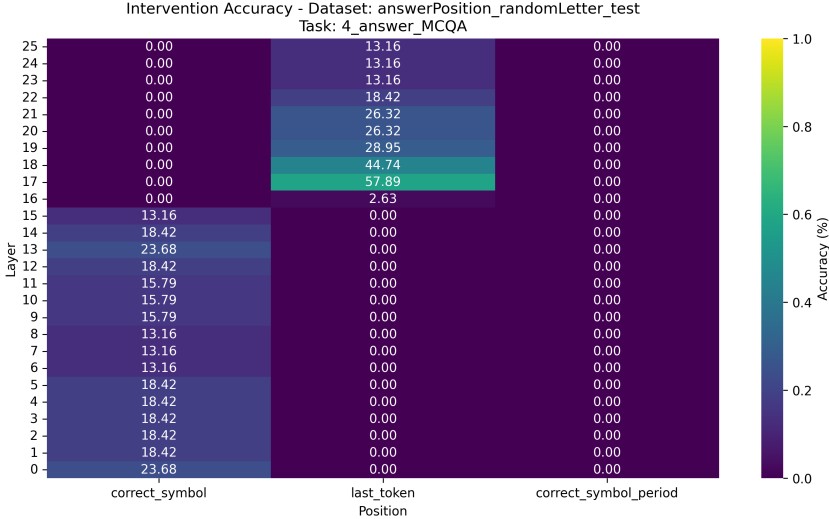

*Figure 8.* $X_{\text{Order}}$ variable alignment results with full residual vector patching.

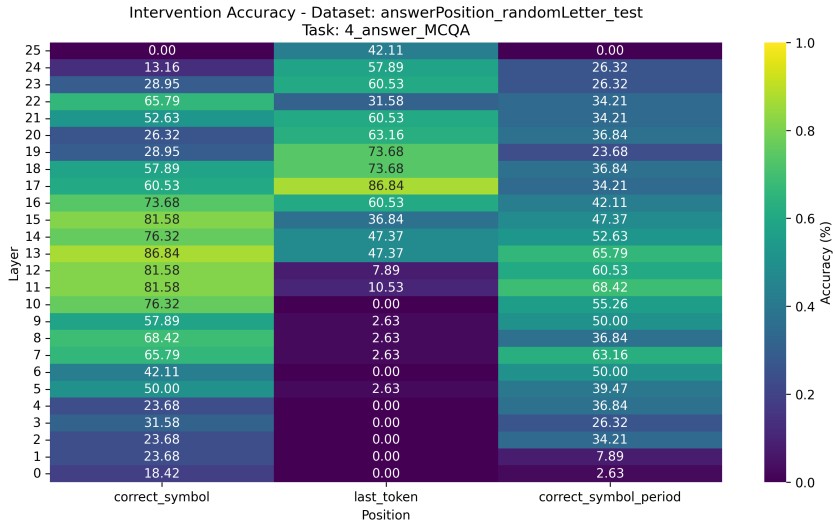

*Figure 9.* $X_{\text{Order}}$ variable alignment results using the subspace identified using the DAS method.

Figures 10 and 11 illustrate the alignment results of the $O_{\text{Answer}}$ variable in the Gemma model, using the full vector patching and DAS methods, respectively. Both results indicate that the $O_{\text{Answer}}$ variable aligns in the later layers—i.e., after the $X_{\text{Order}}$ variable has been established—which is consistent with our hypothesized high-level causal model, where the $O_{\text{Answer}}$ variable derives its information from the $X_{\text{Order}}$ variable.

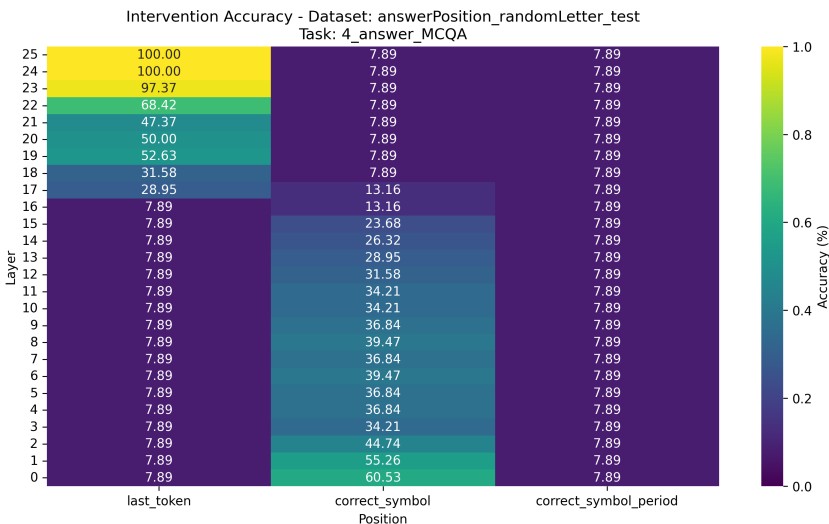

*Figure 10.* $O_{\text{Answer}}$ variable alignment results with full vector patching.

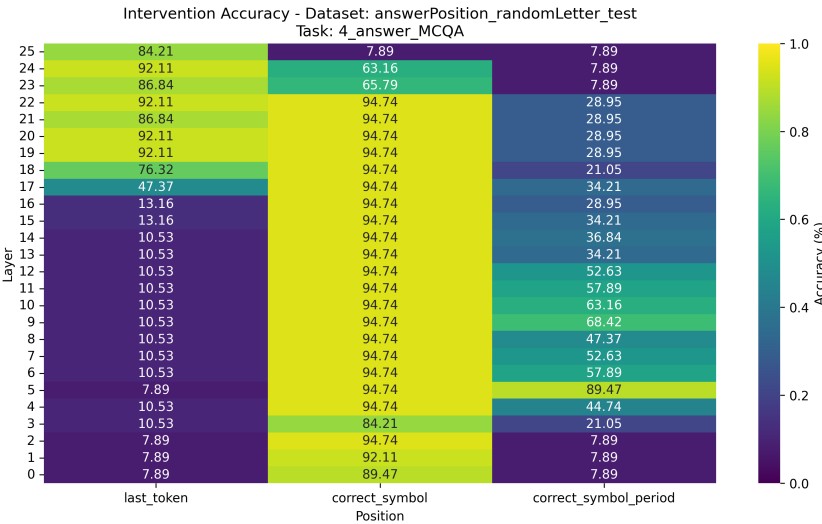

*Figure 11.* $O_{\text{Answer}}$ variable alignment results using the subspace identified using the DAS method.

### G.5.2. ARITHMETIC

We define the causal model for arithmetic (addition) in Figure 14. Unlike the causal models for MCQA and ARC, the addition causal model involves multiple variables, as illustrated in Figure 14a. First, the units and tens digits of both addends are parsed. The unit digits are then added together to determine the units digit of the result, as well as whether a carry is generated. Next, the tens digits of both addends—along with the carry, if any—are summed to compute the tens digit of the result. This step also helps determine whether the result includes a hundred's digit.

Consider the example shown in Figure 14b, namely $57 + 66$. The causal model begins by parsing the addends to identify

their respective tens and units digits. It first adds the unit digits, 7 and 6, determining that the units digit of the result is 3, and that a carry is generated. Next, it adds the tens digits of both addends along with the carry, concluding that the tens digit of the result is 3. Using these same three values—the tens digits and the carry—the model also determines that the result includes a hundreds digit, which is 1.

We evaluate the hypothesized causal model by conducting an interchange intervention experiment to align the $X_{\text{Carry}}$ variable. The counterfactual examples are designed such that they introduce a carry when the original does not, and remove it when the original includes one. Figures 12 and 13 present the alignment results in the Llama model using full vector patching and the DAS method, respectively. Together, these results suggest that the $X_{\text{Carry}}$ information emerges at the last token position during the middle layers of the model.

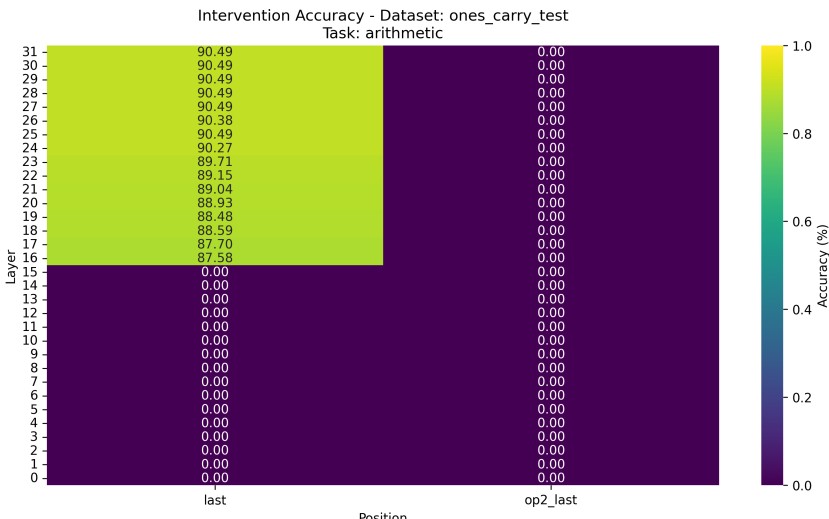

*Figure 12.* $X_{\text{Carry}}$ variable alignment results in Llama with full vector patching.

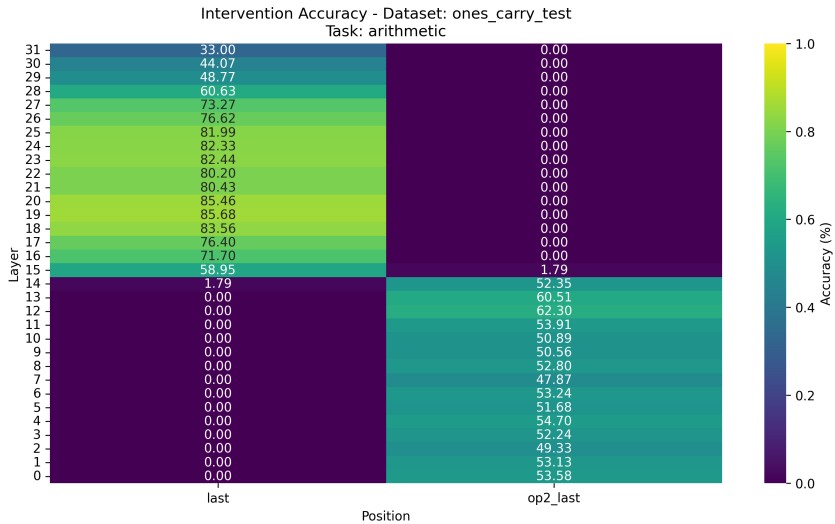

*Figure 13.* $X_{\text{Carry}}$ variable alignment results in Llama using the DAS method.

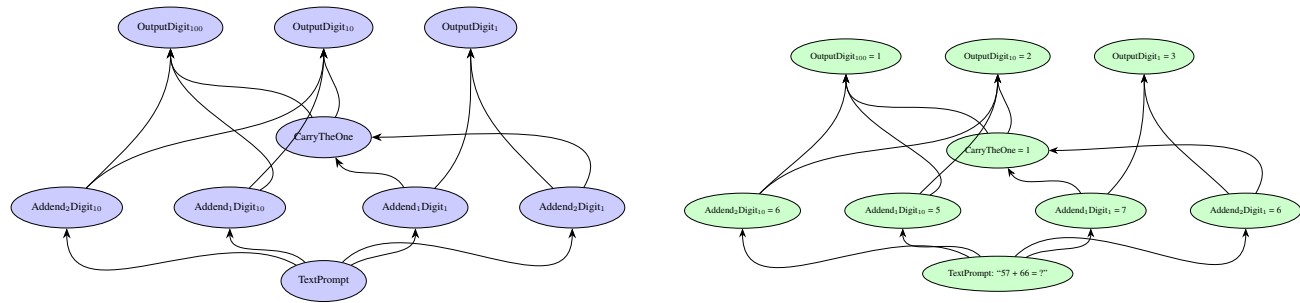

(a) General causal model for two-digit addition          (b) Specific example: 57 + 66 = 123

*Figure 14.* Causal model for two-digit addition arithmetic. The model processes addition through a series of interdependent mechanisms: (1) The 1's digits from both addends directly influence the output 1's digit through modular addition. (2) When the sum of 1's digits exceeds 9, the CarryTheOne variable becomes 1, otherwise it's 0. (3) The 10's digit of the result is determined by three inputs: the 10's digits of both addends and the CarryTheOne value. (4) The 100's digit is causally influenced by both 10's digits of the addends and the carry operation—it becomes 1 only when the sum of 10's digits plus any carried value exceeds 9.

### G.5.3. RAVEL

*Figure 15.* The RAVEL task with variables for the country $A_{Country}$, continent $A_{Cont}$, and language $A_{Lang}$ of a city.

| | RAVEL | | | | | |
| --- | --- | --- | --- | --- | --- | --- |
| | Gemma-2 | | | Llama-3.1 | | |
| Method | $A_{Cont}$ | $A_{Country}$ | $A_{Lang}$ | $A_{Cont}$ | $A_{Country}$ | $A_{Lang}$ |
| DAS | 75 (**85**) | 57 (**67**) | 62 (**70**) | 75 (**83**) | 58 (**64**) | 63 (**70**) |
| DBM | 66 (**71**) | 53 (**65**) | 54 (**58**) | 68 (**80**) | 53 (**59**) | 58 (**64**) |
| +PCA | 63 (**70**) | 47 (**53**) | 50 (**56**) | 62 (**74**) | 48 (**54**) | 53 (**57**) |
| +SAE | 64 (**72**) | 49 (**56**) | 53 (**59**) | 64 (**72**) | 50 (**57**) | 55 (**57**) |
| Full Vector | 48 (**62**) | 49 (**57**) | 45 (**56**) | 53 (**62**) | 47 (**53**) | 47 (**57**) |

The Resolving Attribute–Value Entanglements in Language Models (RAVEL) benchmark (Huang et al., 2024a) evaluates methods for isolating *attributes* of an *entity*. We include the split of RAVEL for disentangling the country, continent, and language attributes of cities. The prompts are queries about a certain attribute, e.g., *Paris is on the continent of*.

**Behavioral Performance.** The Llama model achieves 66.5% accuracy on the task and the Gemma model achieves 70.5% accuracy on the task. The dataset contains many, many cities, so considering that there are many countries and languages to choose from, this is good performance. We filter out failure cases.

**Causal model.** The causal model has a text input variable $T$, three attribute variables $A_{Country}$, $A_{Cont}$, and $A_{Lang}$ for the city in the prompt, a queried attribute variable $A_{Query}$, and an output variable $O$ that retrieves the value of the attribute variable corresponding to the queried attribute.

**Counterfactuals.** Half of the counterfactual prompts are prompts that query a different attribute from the base prompt. The other half are random sentences from Wikipedia containing the city. Interventions on $A_{Country}$, $A_{Cont}$, and $A_{Lang}$ will only change the output if the queried attribute matches the intervened variable. When evaluating each variable, we balance the base prompts such that half of the prompts query the intervened attribute, which enforces the balance between interventions that should change the output and interventions that shouldn't change the output.

**Results.** For each baseline, we target the last token and the last token of the city entity in the prompt. We generally see evidence that the attributes can be localized and disentangled (Figure 15; App. G.5.3), though the Llama-3 results are weaker. Because the dataset requires balancing causing an attribute to change with not changing the other attributes, the full vector baseline entirely fails completely.

**Further Details** We define the causal model for the RAVEL task in Figure 16. The model takes an input prompt that queries the value of an attribute of an entity (e.g., the continent of a city) and outputs the correct answer. As illustrated in Figure 16a, the causal model first parses the input prompt $T$ to extract two input variables: the *entity* and the *queried attribute* $A_{Query}$. The entity refers to a city, which is associated with three attribute variables: $A_{Cont}$, $A_{Country}$, and $A_{Lang}$. The model then identifies the value of each attribute variable for the given entity. Lastly, the answer variable $O$ selects the appropriate value based on the queried attribute. For example, given the prompt "Paris is in the continent of" in Figure 16b, the model first identifies the entity "Paris" and the queried attribute "continent." It then uses its internal knowledge to retrieve the continent, country, and language associated with the city of Paris. Lastly, since the queried attribute is "continent," the model outputs "Europe."

We conduct interchange intervention experiments on each of the attribute variables, $A_{Cont}$, $A_{Country}$, and $A_{Lang}$, to align the language model's internal representations with the hypothesized causal model. We use two types of counterfactuals: the *attribute counterfactual*, which alters the queried attribute in the base prompt, and the *Wikipedia counterfactual*, which is a freeform sentence from Wikipedia about the entity city. If the MI method successfully isolates the target attribute, then the intervention should cause the language model to output the corresponding attribute value for the entity city in the counterfactual.

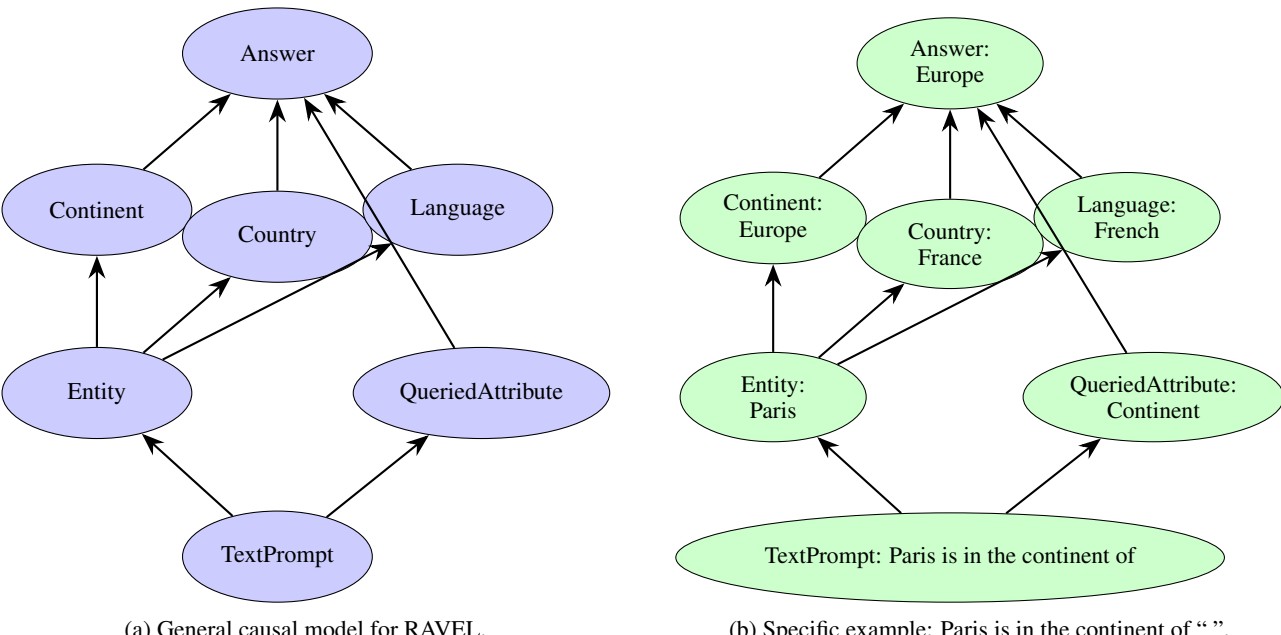

(a) General causal model for RAVEL.  (b) Specific example: Paris is in the continent of " ".

*Figure 16.* Causal model for the RAVEL task. Given a prompt querying an attribute of a city entity, the model extracts the entity and queried attribute as input variables, then identifies the values of the $A_{Cont}$, $A_{Country}$, and $A_{Lang}$ attributes for the entity. Lastly, it decides which value to output based on the queried attribute.

**Results.** We present representative results from the set of intervention experiments. Specifically, we focus on the Gemma-2 2B model and target the $A_{Country}$ variable. Figure 17 shows the Interchange Intervention Accuracy (IIA) at each layer using the *attribute counterfactual*, comparing the baseline (Full Vector) and the best-performing method (DAS). Half of this counterfactual dataset consists of prompts querying the $A_{Country}$ variable, while the other half queries the $A_{Cont}$ or $A_{Lang}$ variables. Successfully isolating the $A_{Country}$ variable requires the method to change only the value of $A_{Country}$ while preserving the values of $A_{Cont}$ and $A_{Lang}$. Since the Full Vector swaps out all features, it performs poorly in the last token position in the later layers. In contrast, the featurizer learned by DAS achieves high IIA in layers 18 and 19.

We also evaluate performance using the *Wikipedia counterfactual*, which presents a more challenging case: the counterfactual prompt is a freeform sentence that does not query an attribute, and thus would reduce false positives where the counterfactual answer coincides with the base prompt. In Figure 18, we observe similar but generally lower IIA patterns compared to the *attribute counterfactual*. The Full Vector baseline fails to isolate the variable, with IIA in both token positions consistently around or below 50. For DAS, IIA increases from the early to mid layers, with meaningful signals emerging in the entity's

last token position in the mid layers. It is worth noting that the $A_{Country}$ variable may be more difficult to disentangle than others ([Huang et al., 2024a](#)), and all five methods we evaluate exhibit lower IIA on $A_{Country}$ compared to the other two variables.

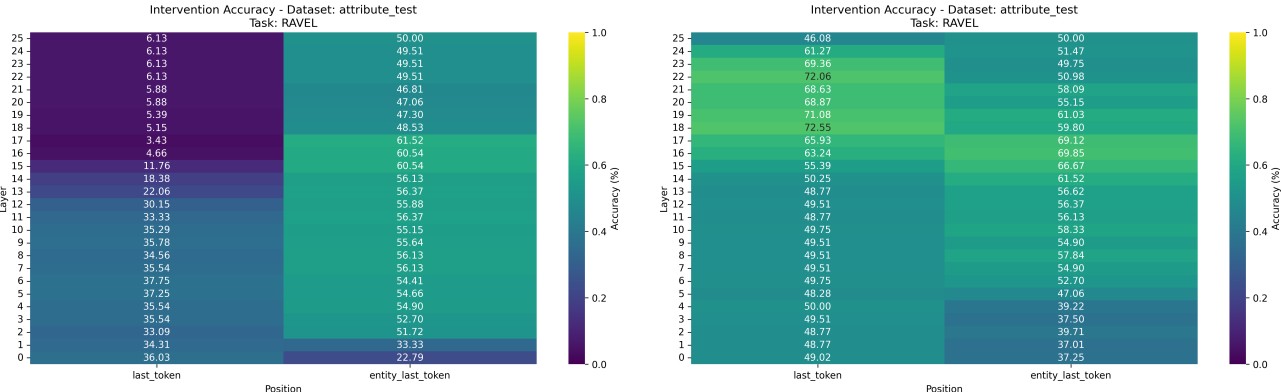

(a) IIA using the identity featurizer (Full Vector).   (b) IIA using Distributed Alignment Search to learn a featurizer.

*Figure 17.* Interchange Intervention Accuracy (IIA) at each layer of Gemma-2 2B, when targeting the $A_{Country}$ variable using the *attribute counterfactual*. The Full Vector baseline fails to isolate the target variable, whereas DAS achieves high accuracy in the mid layers.

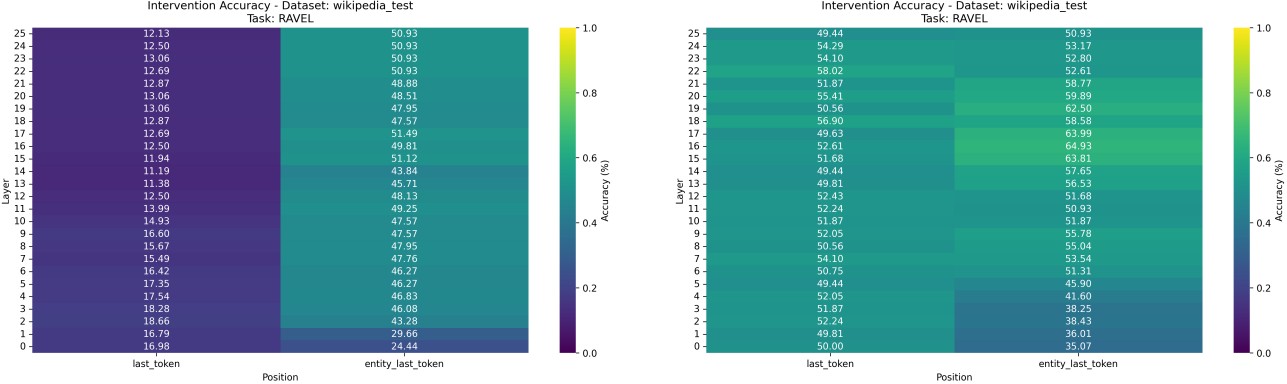

(a) IIA using the identity featurizer (Full Vector).   (b) IIA using Distributed Alignment Search to learn a featurizer.

*Figure 18.* Interchange Intervention Accuracy (IIA) at each layer of Gemma-2 2B when targeting the $A_{Country}$ variable using the *Wikipedia counterfactual*. This counterfactual presents a more challenging setting. The Full Vector baseline fails to disentangle the attributes, while DAS identifies meaningful signal in the mid layers.

### G.5.4. INDIRECT OBJECT INDENTIFICATION

The indirect object identification (IOI) task is a natural language task which consists of sentences like "When Mary and John went to the store, John gave a drink to", and evaluates for the model completion, 'Mary'. The task is linguistically fundamental and has an interpretable algorithm: given two names in a sentence, predict the name that isn't the subject of the last clause.

A sentence in IOI has two parts: a beginning clause that depends on the rest of the sentence, like "When Mary and John went to the store," and a main clause, like "John gave a bottle of milk to Mary." The beginning clause introduces the indirect object (IO), 'Mary', and the subject (S), 'John'. The main clause mentions the subject again, and in every IOI example, the subject gives something to the IO. The goal of the IOI task is to predict the last word of the sentence, which should be the IO. Our high-level model predicts the *logit difference* resulting from subtracting indirect object name logit from the the subject name logit. When this difference is positive, the model is more likely to predict the subject than the indirect object.

(Wang et al., 2023) identify that heads 3 and 9 in layer 7 (7.3 and 7.9) and heads 6 and 10 in layer 8 (8.6 and 8.10) all reduce the likelihood of the model outputting the subject, dubbing these heads "S-inhibition heads". These heads help the model output the indirect object to solve the task. Specifically, Wang et al. (2023) found that S-Inhibition Heads use two types of signals. The first is the token signal, which carries the token identity of the subject, while the second is the position signal, which carries information about the position of subject, i.e., first or second. These information signals carried by the S-Inhibition heads inform other components in the model to avoid the token and position of the subject.

To identify these two signals, Wang et al. (2023) perform interchange interchange interventions that manipulate each signal separately. The IO↔S1-Flip counterfactual inverts the position of the subject while keeping the token the same. The IO↔S2-Flip inverts the token of the subject while keeping the position the same. The IO↔S1-Flip+IO↔S2-Flip inverts the position and token of the subject.

After an interchange intervention is performed on all four heads with one of these counterfactuals, Wang et al. (2023) consider the token/position to have value 1 if unchanged and value -1 if inverted.[13] Using these binary signals as inputs and the logit difference between indirect object and subject as outputs, they perform a regression and find that $2.31\texttt{PositionSignal} + 0.99\texttt{TokenSignal}$ is the best predictive model of logit difference. On our dataset, we perform the same intervention experiments and fit a linear model, finding that $0.069 + 2.018\texttt{PositionSignal} + 0.687\texttt{TokenSignal}$ is the best predictor of logit difference.

Wang et al. (2023) never conducted experiments attempting to disentangle the position and token signals, and this is what we do here.

**Dataset.** Our dataset consists of train, validation, and test splits. The train, test, and private test sets contain 10000 examples each, as well as 10000 examples for 8 corresponding counterfactuals as shown in Table 6.

**Causal Model.** We define the causal model for indirect object identification in Figures 19. The variables $S_{\text{Tok}}$ and $S_{\text{Pos}}$ are derived from the original prompt and without intervention, they will always take on the value of the subject token identity and subject position, respectively. These variables are not equivalent to the binary signals $\texttt{TokenSignal}$ and $\texttt{PositionSignal}$. Instead, the output $O_{\text{LogDiff}}$ variable compares the subject token and subject position to the original input to determine whether the token and position is inverted and run the linear model accordingly.

Without an intervention, the $S_{\text{Tok}}$ and $S_{\text{Pos}}$ variables will always match the text input, and the logit diff is predicted to be $0.069 + 2.018 + 0.687$. When only $S_{\text{Tok}}$ is intervened on, the logit diff is predicted to be $0.069 + 2.018 - 0.687$. When only $S_{\text{Pos}}$ is intervened on, the logit diff is predicted to be $0.069 - 2.018 + 0.687$. If both are intervened on, then the logit diff is predicted to be $0.069 - 2.018 - 0.687$

**Full Vector Brute-Force Search** We conduct a brute-force search by aligning each of the two causal variables with every possible subset of the four S-inhibition heads. In Table 18) we report the mean-squared error for the alignments. In the main text, we report the alignment of $S_{\text{Pos}}$ to heads 7.3, 7.9, and 8.6 and $S_{\text{Tok}}$ to the head 8.10.

---

[13]They also included a value of 0 for the token signal of a random new token, but we leave out this condition for our experiments.

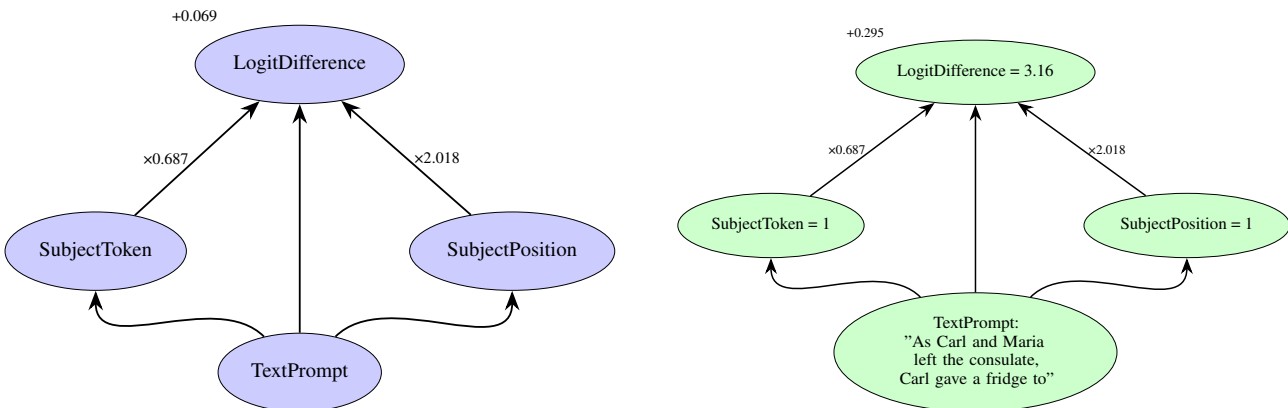

(a) General causal model for indirect object identification

(b) Specific example showing prediction for "Carl"

*Figure 19.* Causal model for indirect object identification. The TextPrompt is processed to extract the subject token $S_{\text{Tok}}$ and the subject position $S_{\text{Pos}}$. The output variable mechanism (1) compares the token and position to the input, and determines whether the token and position were inverted or the same and sets `PositionSignal` and `TokenSignal` to 1 and -1 accordingly and (2) computes $0.069 + 0.687\texttt{TokenSignal} + 2.018\texttt{PositionSignal}$, with position having a stronger influence than token identity.

*Table 18.* Mean Squared Error (MSE) for aligning $S_{\text{Pos}}$ and $S_{\text{Tok}}$ to each subset of heads.

| Heads | MSE ($S_{\textbf{Pos}}$) | MSE ($S_{\textbf{Tok}}$) |
|---|---|---|
| **((7, 3), (7, 9), (8, 6))** | **2.45** | 6.83 |
| ((7, 3), (7, 9), (8, 10)) | 3.08 | 6.79 |
| ((7, 3), (7, 9)) | 5.42 | 3.28 |
| ((7, 3), (8, 6), (8, 10)) | 2.52 | 8.08 |
| ((7, 3), (8, 6)) | 4.17 | 4.82 |
| ((7, 3), (8, 10)) | 4.92 | 3.51 |
| ((7, 3)) | 12.03 | 4.07 |
| ((7, 9), (8, 6), (8, 10)) | 2.90 | 10.20 |
| ((7, 9), (8, 6)) | 3.15 | 5.05 |
| ((7, 9), (8, 10)) | 3.63 | 4.33 |
| ((7, 9)) | 7.90 | 2.80 |
| ((8, 6), (8, 10)) | 3.00 | 5.71 |
| ((8, 6)) | 5.81 | 4.21 |
| **((8, 10))** | 7.28 | **2.82** |

