# OpenReview forum: "MIB: A Mechanistic Interpretability Benchmark"
_ICML.cc/2025/Conference — ICML 2025 poster_

### Official Review · Reviewer_NtjL · 2025-03-13

**Overall Recommendation:** 4

**Summary:**

This paper proposes a benchmark, called MIB, to evaluate whether the interpretability algorithm precisely and concisely recovers relevant
causal pathways or specific causal variables. MIB includes two tasks: 1) circuit localization which identifies important connections in the model to perform a task and 2) causal variable localization which compares different ways to project hidden features to find causal mediators. MIB assesses on many common tasks used in the field, standardizing the evaluation and providing insights into the capability of MI methods.

**Claims And Evidence:**

The claims in the paper are supported by the experiments.

**Essential References Not Discussed:**

This benchmark does not discuss other metrics in prior work.

[1] Interpretability in the Wild: a Circuit for Indirect Object Identification in GPT-2 small, ICLR 2023

[2] Sparse Feature Circuits: Discovering and Editing Interpretable Causal Graphs in Language Models, ICLR 2025

[3] Scaling and evaluating sparse autoencoders, ICLR 2025

**Experimental Designs Or Analyses:**

Please see the comment in the Evaluation section.

**Methods And Evaluation Criteria:**

- Both tracks in this benchmark only evaluate the faithfulness of the explanation; however, there are other criteria as well, such as completeness, minimality and human-interpretability [1,2]
- Previous work [3] criticizes the faithfulness in the circuit localization track and suggests considering the relative amount of pretraining compute needed to achieve comparable performance. Could you include this metric in the benchmark or discuss its utility compared to the faithfulness metric?
- In the causal variable localization track, the IOI task uses a different metric than other tasks and this metric is not presented in the text. Could you explain why we should use different metrics here? Also, I don't understand how we can calculate the logit of the high-level causal model in the IOI task. How do you compute this metric or could you point to the description in the manuscript if I miss it by any chance?

[1] Interpretability in the Wild: a Circuit for Indirect Object Identification in GPT-2 small, ICLR 2023

[2] Sparse Feature Circuits: Discovering and Editing Interpretable Causal Graphs in Language Models, ICLR 2025

[3] Scaling and evaluating sparse autoencoders, ICLR 2025

**Other Comments Or Suggestions:**

Can we, and should we, study the interaction of circuit localization and causal variable localization? More specifically, if we find a circuit with different types of features, do the observations in Sec 3 still hold?

**Other Strengths And Weaknesses:**

- The presentation could be improved. For example, it'd be helpful if you could explain why the subgraph performance ratio locates components with a positive effect and the subgraph behavioral distance locates components with any strong effect.
- The notation is confusing and not explained. For example, in the causal variable track, $\mathcal{C}$ in interchange intervention overlaps with the notation of the circuit in the previous section, while later in the formula interchange intervention uses $\mathcal{A}$, which represents the high-level causal model.

**Questions For Authors:**

Please see the question above.

**Relation To Broader Scientific Literature:**

This work provides a benchmark for developing MI methods.

**Theoretical Claims:**

There is no theoretical result.

---

> ### Author Rebuttal · Authors · 2025-03-31
>
> Thank you for your valuable comments. We have a detailed plan to address your points about readability and presentation. If accepted, will we use the additional page to incorporate this material to the main text.
>
> > We use faithfulness, but should also discuss completeness, minimality, and human-interpretability
>
> Thanks for this suggestion. One reason we propose using the area under the faithfulness curve is because this captures both **minimality** and **faithfulness** at the same time: methods that are better at locating the most important causal dependencies with fewer components will have higher faithfulness at smaller circuit sizes, thus increasing the area under the curve. Our metrics thus reward methods that are good at locating minimal circuits.
>
> Measuring **completeness** is generally not computationally feasible. Wang et al. (2023) performed a detailed manual analysis on a single task; this allowed them to discover clusters of components with similar functional roles. Their notion of completeness involves ablating subsets of a particular functional cluster. In automatic circuit discovery settings, we don't know what the ground-truth clusters should be, and would thus need to enumerate all possible combinations of components (an exponential-time search). Thus, most work in this field does not measure completeness. Marks et al. (2024) propose ablating only the circuit to measure completeness, but it is much easier to destroy performance than to recover it (i.e., low scores when ablating the circuit don't necessarily imply that we have found all important causal dependencies). As for **human-interpretability**, we see this as the role of featurization methods, rather than circuit localization methods; higher performance in the causal variable localization track implies that the method is better at locating human-interpretable concepts in models. We will discuss these ideas and challenges explicitly in the final version; thanks!
>
> > "Previous work criticizes faithfulness on circuits and suggests considering the relative amount of pretraining compute needed to achieve comparable performance...?"
>
> We're a bit confused by this feedback; could you clarify? Gao et al. doesn't criticize faithfulness, and they compare training compute for SAEs. Circuit discovery typically does not involve training.  More broadly, typical language modeling benchmarks use model size (akin to our minimality metric) and model performance (akin to our faithfulness metric), but tend not to compare models based on compute.
>
> > "In the causal variable localization track, the IOI task uses a different metric, Why this metric?"
>
> Because the high-level causal model for the IOI task predicts the output logits of the model (rather than a specific behavior), we cannot use accuracy. Instead, we measure the squared error between the high-level model logits and the actual language model logits. We have clarified this in the new draft. Please see L424-429 for how we compute the logit $O$ of the high-level causal model using a linear model over binary variables.
>
> > We don't discuss metrics used in IOI paper, sparse feature circuits paper, and scaling SAEs paper
>
> Please see responses above for a discussion of the metrics proposed by these papers. We will add this discussion to the revision.
>
> >  “why the subgraph performance ratio locates components with a positive effect ...” and “the causal variable track, in interchange intervention overlaps with the notation of the circuit...”
>
> Please see the response to RnEi for an intuitive summary of the circuit localization metrics, and a description of how we will revise notation and presentation for clarity.
>
> > “notation is confusing… $\mathcal{C}$ in interchange intervention overlaps with the notation of the circuit in the previous section, while later in the formula interchange intervention uses $\mathcal{A}$, which represents the high-level causal model”
>
> Good point; we have standardized notation across tracks, and added the table for notation seen here: https://imgur.com/a/m4DDNH2. To avoid overloading notation, we now use $\mathcal{H}$ for a high-level causal model, and $\mathcal{C}$ for a circuit. $\mathcal{C}$ refers to a part of the computation graph, whereas $\mathcal{H}$ is an abstraction that does not necessarily map cleanly to the computation graph. We also made the notation for interchange interventions more transparent, with $\leftarrow$ indicating intervention.
>
> > What about the intersection of the causal variable localization and circuit localization tracks?
>
> We agree completely! Our evaluations focus primarily on localization *or* featurization, but future work should consider their intersection. We felt that good metrics for each were necessary before one could consider evaluating the two jointly. Future work could consider metrics that are compatible with circuits built on sparse features, pre-located causal variables, or neuron clusters rather than individual neurons or submodules.

---

> > ### Comment · Reviewer_NtjL · 2025-04-04
> >
> > Thank you for your clarification. I'd like to discuss some points in the review
> >
> > **Q1:** I agree with the argument on minimality and the AUC metric. I think it should be highlighted as the motivation of the proposed metric.
> >
> > About completeness, it'd be more convincing if you could justify `it is much easier to destroy performance than to recover it ` by showing that the ablation test does not provide any statistically significant evidence that the MI method finds all the explanations.
> >
> > About human-interpretability, I don't totally agree that it could be expressed by higher performance in the causal variable localization track. The faithfulness metric shows how the features align with causal variables in the **predefined hypothesis posed by humans**. There could be the case that the model implements another algorithm, which is still human-understandable yet different from the hypothesis. For example, although it's counterintuitive, LLMs could perform arithmetic by converting the numbers to base 2 and performing carry-the-one. That said, I believe that it's challenging to design such a test without human-in-the-loop, and it should be acknowledged as a current limitation.
> >
> > **Q2:** Sorry for the wording; I didn't mean Gao et al. criticize the faithfulness but the way it's computed, i.e., by the drop in the loss, with the similar argument as in your rebuttal of Q1. More particularly, in Sec. 4.1 in their paper, they suggest considering the relative amount of pretraining compute needed to train a language model of comparable downstream loss. Although it could be resource-intensive, I believe that for the goal of a benchmark, where we are not limited by any constraint and want a complete assessment of the method, a more reliable metric is more useful.
> >
> > The responses to other questions addressed my concerns.

---

> > > ### Author Response · Authors · 2025-04-07
> > >
> > > **Q1:** Agreed, we will emphasize this point!
> > >
> > > On **completeness**: the issue with significance testing is that any meaningful test would require access to the ground-truth set of causally influential components. For realistic models, this ground-truth set cannot be tractably computed without significant manual effort. One could likely achieve the same low performance by ablating an entire circuit or just 75% of it, so it's hard to derive a tractable and automatic signal. Importantly, **our InterpBench model addresses this limitation**! Its AUROC metric captures completeness, minimality, and faithfulness. We will add a note about completeness for non-InterpBench models to the limitations.
> > >
> > > On **human-interpretability**: we believe your comment is compatible with ours! Our faithfulness metric captures the extent to which the causal variable—not the entire high-level model—aligns with the representation. The high-level model may differ from than the hypothesis, but it would still be possible to modify the model's behavior in a predictable way, and this will be reflected in the scores. We will follow your suggestion and discuss the implications of different possible high-level models in the limitations; thank you!
> > >
> > > **Q2:** Thanks for following up. In circuit localization, no method involves training (except UGS). Most methods involve gradient attributions or inference-time interventions, which are non-parametric and use little compute. In the causal variable track, only some of the methods are parametric. Given this methodological diversity, any notion of "training compute vs. performance" would only be applicable to a small subset of the methods, and runtime metrics could be misleading if applied to non-parametric and parametric methods.
> > >
> > > Given the above discussion, adding such a metric to the core benchmark would be problematic. That said, because circuit discovery methods often don't require training, it's often possible to estimate the number of forward and backward passes needed to discover a circuit:
> > > | Method | Num. Passes |
> > > |-----------|-----------|
> > > | AP | $O(d)$ |
> > > | AP-IG-inputs | $O(d \cdot k)$ |
> > > | AP-IG-activations | $O(d \cdot k \cdot L)$ |
> > > | ActP | $O(d \cdot N)$ |
> > > | IFR | $O(d)$ |
> > > | UGS | $O(d \cdot e)$ |
> > > Where $d$ is the number of examples used to find the circuit, $k$ is the number of interpolation steps, usually a small number (we use 10), $L$ is the number of layers in the model, $N$ it the number of components in the model—a large number usually in the tens or hundreds of thousands at least—and $e$ is the number of training epochs. Given the good faithfulness and low number of passes needed for AP-IG-inputs, we believe this strikes the best balance between runtime/compute and performance. The very high number of passes needed for ActP makes it often not worthwhile for larger models. IFR performs relatively poorly but requires few passes, whereas UGS is usually somewhere in the middle w.r.t. runtime and performance. While helpful for our baselines, note that number of passes isn't necessarily the main factor in runtime or compute for all circuit discovery methods, so we would hesitate to formalize this as a general metric.
> > >
> > > For causal variable localization, DAS, DBM, and SAE (the parametric methods) have similar runtimes; PCA (a non-parametric but data-driven method) is faster, but tends to perform poorly. None (Full Vector) requires no training, nor any forward/backward passes before evaluation.
> > >
> > > We hope this addresses your concerns. Please let us know if you have remaining questions or feedback!

---

### Official Review · Reviewer_RnEi · 2025-03-14

**Overall Recommendation:** 3

**Summary:**

The authors proposed a benchmark dataset for Mechanistic Interpretability (MI). The dataset consists of four tasks: 1) Indirect Object Identification (IOI), 2) Arithmetic with two digits, 3) Multiple-Choice Question Answering (MCQA), and 4) AI2 Reasoning Challenge (ARC). The goal of the benchmark is to test for circuit localization and causal variable localization. The authors include metrics for each of the goals and test the performance of several baselines on each task and each goal for different language models.

**Claims And Evidence:**

The submission doesn’t make many theoretical or empirical claims, it does make observations about the current state of MI which is supported by their experiments.

**Essential References Not Discussed:**

Not that I know of.

**Experimental Designs Or Analyses:**

Yes, I checked the validity of the experimental design and it seems serious to me. They tried several baselines and several models on all of their tasks they proposed.

**Methods And Evaluation Criteria:**

It is hard to tell whether the evaluation criteria (the metrics they use to evaluate the baselines) are good or not, I think the presentation of the paper is far from ideal, at least from someone who does not already know most of the MI literature.

**Other Comments Or Suggestions:**

See weaknesses.

Would be very nice to include graphs for the causal relations described in text in 4.2. In the end, if I understand correctly, these causal graphs are fixed for the tasks at hand.

**Other Strengths And Weaknesses:**

Strengths:
- As mentioned above, I think there is a lot of value in establishing a common benchmark for MI given the attention it is currently getting from the research community.
- I appreciate the environment the authors are trying to build around the benchmark. For example, including it in Hugging Face, or even providing compute to test the user models.
- I find valuable both the diversity of the tasks/tracks and the possibility of adding more tasks as the research and models evolve.
- I can see the authors put some thought into the experiments and the baselines they tested.

Weaknesses:
- My biggest problem with the paper is its readability. For theoretically oriented people, or even people who are not already deeply in the MI and NLP research it is very difficult to read. Let me give a couple of examples.
    - In 3.1 when they describe the circuit metrics, they mention that faithfulness is what is commonly used. Then they say what are the goals of faithfulness and argue they need to separate these goals into two metrics for each of the goals, the notation which I already find a bit confusing (F_{=}, F_{+}), and then with no justification they propose a proxy to estimate these metrics. I was lost there.
    - d_{model} in line 237 is not defined, which makes the next line even more difficult to parse. The whole sentence reads:
“Including one neuron of d_{model} in submodule u can be conceptualized as including all outgoing edges from u to 1/d_{model} of the degree they would have been compared to including all neurons in u.”
    - The “faithfulness metrics.” paragraph on section 4 suffers from the same, although to a lesser extent, I would say.

**Questions For Authors:**

If the authors can describe in an actionable way how they will address the readability of the paper I am happy to increase my score.

**Relation To Broader Scientific Literature:**

I find the paper to be very valuable in relation to the literature. Mechanistic interpretability is very popular now and having a common benchmark is definitely important.

**Theoretical Claims:**

NA

---

> ### Author Rebuttal · Authors · 2025-03-31
>
> Thank you for appreciating the value of our paper’s contribution and the validity of our experimental designs. We note your points about readability and presentation, and appreciate your willingness to reconsider your score on the basis of correcting them. We will make a number of changes to improve readability and presentation. If accepted, will we use the additional page given us to incorporate the following material to the main text and increase the readability of the paper.
>
> Plans to improve readability and presentation
> ===
>
> **”Sec 3.1….argue they need to separate these goals into two metrics for each of the goals, the notation which I already find a bit confusing (F_{=}, F_{+}), and then with no justification they propose a proxy to estimate these metrics. I was lost there”**
>
> We will add some more intuition behind the metrics in the final version of the paper. We realized that the $F_+$ and $F_=$ names were confusing because they are named after the ideal values they should take, rather than what they actually measure. To be more transparent, we have renamed $F_+$ to the **circuit performance ratio** (CPR), and $F_=$ to the **circuit-model distance** (CMD). In short, CPR measures how the circuit performs on the task metric $m$ as a ratio w.r.t. the full model’s performance, while CMD measures how closely the circuit replicates the full model’s input-output behavior. One should aim to maximize CPR if the goal is to find the best-performing circuit; one should aim to minimize CMD if the goal is to find a circuit that concisely implements the full model's behavior.
>
> Both of these metrics are defined using integrals. It is impossible to compute these integrals in exact form without infinite samples. The trapezoidal rule (i.e., Riemann sum) is an established way of approximating integrals given a few values of the function—this is the proxy we allude to. In the revision, we will clarify by not using different names for the exact and empirical definitions. We will instead define CPR and CMD in exact form, and then simply say how we measure them in practice.
>
>
> **”Would be very nice to include graphs for the causal relations described in text in 4.2. In the end, if I understand correctly, these causal graphs are fixed for the tasks at hand.”**
>
> Yes, the causal graphs are fixed for the task at hand. We added a more comprehensive description of the causal variable localization track that includes causal models for each task and figures that illustrate the concepts and terminology. For example: https://imgur.com/a/6mgZfj2.
>
> **"$d_{model}$ in line 237 is not defined, which makes the next line even more difficult to parse"**
>
> Thanks for catching this! We will add the following clarification: "...where $d_{\text{model}}$ is the model's hidden size, or the number of neurons in the activation vector output of each layer."
>
> Beyond these specific points you raised, we have standardized notation across tracks, and added a table of notation, which you may view here: https://imgur.com/a/m4DDNH2. To avoid overloading notation, we have now modified the text so that $\mathcal{H}$ is a high-level causal model and $\mathcal{C}$ is a circuit. $\mathcal{C}$ refers to a part of the computation graph, whereas $\mathcal{H}$ is an abstraction that does not necessarily map cleanly to the computation graph. We also made the notation for interchange interventions more transparent with the $\leftarrow$ indicating intervention.
>
> Other comments
> ===
>
> > "Submission doesn't make any theoretical or empirical claims"
>
> Please see our response to reviewer bp8X on scientific novelty.
>
> > "It is hard to tell whether the evaluation criteria (the metrics they use to evaluate the baselines) are good or not"
>
> A sanity check for our evaluation criteria is that they recover known findings. In the circuit localization track, attribution patching with integrated gradients is significantly better than attribution patching, in line with prior work [1,2]. In the causal variable localization track, supervised methods outperform unsupervised methods, as expected, and as found in very recent work [3].
>
> References
> ===
> [1] Hanna et al. (2024). "Have Faith in Faithfulness: Going Beyond Circuit Overlap When Finding Model Mechanisms." COLM. https://arxiv.org/abs/2403.17806
>
> [2] Marks et al. (2025). "Sparse Feature Circuits: Discovering and Editing Interpretable Causal Graphs in Language Models." ICLR. https://arxiv.org/abs/2403.19647
>
> [3] Wu et al. (2025). "AxBench: Steering LLMs? Even Simple Baselines Outperform Sparse Autoencoders." arXiv. https://arxiv.org/abs/2501.17148

---

### Official Review · Reviewer_Pv8S · 2025-03-15

**Overall Recommendation:** 3

**Summary:**

Mechanistic Interpretability (MI) research has made significant strides, but it has lacked a consistent way of comparing methods. In this paper, the authors introduce Mechanistic Interpretability Benchmark (MIB), which splits the evaluation into two main tracks: Circuit Localization and Causal Variable Localization. Across four tasks, MIB allows researchers to test how well different MI methods can (1) identify the causal circuit responsible for a given task behavior, and (2) localize specific conceptual variables within a model’s hidden representations. Using MIB, the authors show that attribution patching and mask-optimization approaches perform best in circuit localization, whereas supervised methods outperform unsupervised ones in aligning causal variables.

**Claims And Evidence:**

**Claim 1**: MIB provides a consistent evaluation framework, enabling direct comparison among MI methods.
- Evidence: The authors introduce a publicly hosted leaderboard for each of the two tracks. Researchers can submit their methods to get systematic performance scores, facilitating transparent head-to-head comparisons.

--> Convincing and clear

**Claim 2**: In the circuit localization track, attribution-based patching and mask-optimization methods are most effective.
- Evidence: Using the authors’ proposed faithfulness metrics F+ and F=, EAP-IG-inputs consistently achieves top results across multiple tasks (Tables 2 and 13).

--> Convincing and clear

**Claim 3**: In the causal variable localization track, supervised methods outperform unsupervised methods.
- Evidence: DAS and DBM achieve higher interchange intervention accuracy (faithfulness) across tasks such as MCQA and ARC (Table 3).

--> Convincing and clear

**Essential References Not Discussed:**

No critical reference omissions are apparent.

**Experimental Designs Or Analyses:**

- The authors systematically benchmark multiple known MI methods across four tasks.

**Methods And Evaluation Criteria:**

- The authors isolate two main dimensions of mechanistic interpretability: localizing all edges/nodes that implement a task and localizing a single conceptual variable.
- The faithfulness metrics are well-designed in each track.

**Other Comments Or Suggestions:**

Please refer to questions.

**Other Strengths And Weaknesses:**

**Strengths**:
- Thorough coverage of existing methods and LLM models demonstrates both broad scope and significant differences in method performance.

**Weaknesses**:
- The task selection rationale could be more comprehensively justified. Why exactly these four, and do they ensure coverage for all important MI phenomena?

**Questions For Authors:**

1. Beyond “representative tasks,” what’s the deeper motivation for picking these four tasks specifically, and how can we be sure this set is sufficient to measure general MI progress?
2. Could you clarify how the weighted edge count integrates into the final scoring? Is it directly used in the F+ and F= metrics, or is it just an informative reference for circuit sparsity?

**Relation To Broader Scientific Literature:**

MIB connects to ongoing attempts to standardize mechanistic interpretability evaluations.

**Theoretical Claims:**

There is no rigid theoretical claim.

---

> ### Author Rebuttal · Authors · 2025-03-31
>
> Thank you for your positive assessment of the thoroughness of our experiments, the convincingness and clarity of our claims, and the design of our metrics!
>
> > Regarding task selection rationale and task coverage:
>
> We aimed to strike a balance between having tasks (1) of diverse difficulties and requiring diverse skills that capture the strengths and weaknesses of different models (L91-92), (2) that have and have not been studied in prior mechanistic interpretability work, and (3) that good open-source language models are capable of performing (Table 1; L131-133). To point (1), the four tasks we selected test linguistic/semantic understanding of text (IOI), mathematical reasoning (Arithmetic), scientific knowledge (ARC), copying from context (MCQA), and the ability to answer formatted multiple-choice questions (MCQA and ARC). To point (2), we included tasks that are more widely studied, such as IOI and Arithmetic, to ground the benchmark in existing and ongoing research and to foster buy-in from the mechanistic interpretability community. We purposefully selected ARC and MCQA to push the research community in a particular direction: towards tasks that are represented on real-world leaderboards. Tasks like IOI have dominated the MI literature (with dozens of papers evaluating on this task), but others such as MCQA have only been studied in 2 papers (neither of which analyzed the ARC dataset). ARC in particular is significantly more realistic than the tasks that have been studied to date using circuit discovery methods.
>
> We intend for the benchmark to be a living resource (L433-438) and plan to expand the task list with community buy-in. We did not mean to imply (and thus do not state in the paper) that the four selected tasks are sufficient for understanding *all* LM behaviors. “[E]nsur[ing] coverage for all important MI phenomena” as requested would involve a task list capturing nearly every textual output an LM can produce! Regarding how we can know our task set is “sufficient to measure general MI progress”: MI progress has up until this point been driven by papers running analysis most commonly on one or two datasets, and our task set has already allowed us to draw meaningful comparisons not established in prior work (see response to bp8X for more on this).
>
> > How do weighted edge counts factor into the $F_+$ and $F_=$ metrics?
>
> $F_+$ and $F_=$ (which have been renamed to circuit performance ratio (CPR) and circuit-model distance (CMD), respectively; see response to RnEi) can be conceptualized as integrals of $f$ over the weighted edge counts. If we plot faithfulness $f$ against circuit size (measured via weighted edge count), then $F_+$ (CPR) is simply the area under this curve. The weighted edge count is a way to measure circuit size in a way that enables direct comparisons across edge-level and node-level circuits; it is basically (proportion of nodes in submodule $u$) times (number of edges outgoing from submodule $u$), summed over all $u$.

---

> > ### Comment · Reviewer_Pv8S · 2025-04-07
> >
> > Thank you for your response. My main concerns were addressed in the rebuttal. I also read the other reviewers' comments and the corresponding responses. While I acknowledge the novelty in proposing an evaluation benchmark for Mechanistic Interpretability, as the authors aimed to do, I did not find additional strengths that would warrant a change in my score. Therefore, I will maintain my current score.

---

### Official Review · Reviewer_bp8X · 2025-03-15

**Overall Recommendation:** 3

**Summary:**

The authors introduce a benchmark designed to standardize evaluations of mechanistic interpretability (MI) methods. This benchmark offers consistent evaluation across standardized models, metrics, and intervention datasets, with two public leaderboards tracking method performance. The benchmark is divided into two specialized tracks:  one evaluating circuit localization methods (identifying computational pathways within networks) and one assessing causal variable localization methods (identifying specific features representing causal variables in hidden representations). They evaluate several SOTA MI methods and discover that attribution and mask optimization techniques perform best for circuit localization, while supervised approaches outperform unsupervised methods for causal variable localization. The benchmark provides a systematic framework for comparing different MI approaches, offering valuable guidance for future research directions in mechanistic interpretability.

**Claims And Evidence:**

The author claims to introduce a new standardized benchmark for evaluation of MI method and also introduce 2 leaderboards open to public submissions. The claim are clear and supported by the description in the paper.

**Essential References Not Discussed:**

Essential references are discussed.

**Experimental Designs Or Analyses:**

The authors evaluate their methodologies on a set of existing MI methods. Specifically, they implement and analyze the results of Circuit Localization Methods in Sections 3.2 and 3.3. The implementation details are well explained and clear.
They implement and analyze the results of  Causal Variable Localization Methods in Sections 4.1 and 4.2. They clearly describe implementation details and report results for a wide range of methodologies.

**Methods And Evaluation Criteria:**

The authors evaluate several existing MI methodologies on their proposed benchmark, reporting results and analyzing methods performance over different tasks. The experiments and evaluations for such methodologies are thoroughly described and clearly reported in sections 3.2 and 4.1 (Tables 2 and 3). In alignment with their proposal of a benchmark for evaluating mechanistic interpretability methodologies, the authors provide comprehensive evaluations of existing MI techniques. Their thorough analysis of these comparative results further strengthens the benchmark's utility for the community.

**Other Comments Or Suggestions:**

None

**Other Strengths And Weaknesses:**

As previously noted, the benchmark provides a valuable foundation for the standardized comparison of mechanistic interpretability methodologies. However, its substantial reliance on existing datasets and evaluation metrics raises questions about whether the technical contribution and novelty meet the threshold for publication at this scientific conference despite its clear practical utility. In short, it is a valuable empirical contribution but lacks scientific novelty. Thus, I am inclined towards a weak accept.

**Questions For Authors:**

None

**Relation To Broader Scientific Literature:**

The paper introduces a new benchmark for standardizing the evaluation of mechanistic interpretability (MI) methodologies. The authors effectively consolidate existing benchmarks and metrics (with some modifications) to create a standardized evaluation framework. They provide publicly accessible leaderboards and conduct comprehensive evaluations of current methods against their benchmark.
This contribution offers value to the research community by addressing the fragmentation in evaluation approaches. While the benchmark doesn't introduce novel datasets, its claimed merit lies in establishing consistent evaluation guidelines and maintaining accessible leaderboards. The benchmark's ultimate impact will largely depend on the usability and maintenance of these leaderboards, but it represents a potentially valuable resource for the field despite limited technical novelty.

**Theoretical Claims:**

The authors introduce 2 faithfulness metrics in section 3.1, one circuit size metric in section 3.1, and an interchange intervention accuracy metric in section 4. The mathematical formulations are based on intuitive, simple concepts and are correct.

---

> ### Author Rebuttal · Authors · 2025-03-31
>
> Thank you for your detailed review and for acknowledging the value of our benchmark’s practical utility and contribution to the literature! We’d like to clarify our scientific contributions;  they include 1) new metrics, and 2) new empirical results (L432-435 Col.2), facilitated crucially by 3) our standardization of metrics, datasets, and models.
>
> > Regarding reliance on existing datasets:
>
> While we included tasks that are more widely studied, such as IOI and Arithmetic, to ground the benchmark in existing research, we purposefully selected our tasks to push the research community towards tasks that are represented on real-world leaderboards (such as MCQA and ARC) with varying formats and difficulty levels (L91-92). Crucially, creating new datasets isn’t necessary when existing datasets remain un- or under-studied. Certain tasks such as IOI have dominated the MI literature (with dozens of papers evaluating on this task), but others such as MCQA have only been studied in 2 papers (neither of which analyzed the ARC dataset). ARC in particular is significantly more realistic than the tasks that have been studied to date using circuit discovery methods. Our main contributions were not the datasets, but rather the metrics, the curation of datasets, and the resulting systematic analyses.
>
> Also, the vast majority of MI papers only run analysis on one dataset and/or one model. In contrast, we conduct systematic analyses across five models, four datasets, and several methods for each track. We are also the first to run experiments on ARC. We expand the synthetic MCQA dataset from prior work from 105 to 260 instances, and the Arithmetic dataset to 150k instances (75k used in our experiments), compared to 1200 in prior work.
>
> Finally, we curate counterfactuals to isolate specific information, which may be useful for other mechanistic studies outside of MIB, as the choice of counterfactual is central to making causal claims [1].
>
> We elaborate more on choice of tasks in our response to Pv8S.
>
> > Regarding reliance on existing metrics:
>
> For the circuit localization track, we do not use existing evaluation metrics; in fact, one of our main contributions is proposing new metrics (see first 2 PPs of Section 3.1)! Faithfulness metrics in prior work conflated important model components (both helpful and harmful) with components driving better model behavior. Additionally, previous studies typically measure the quality of a single circuit using a single faithfulness value $f$, rather than measuring the quality of a method via an aggregation over $f$ values (as we do). The weighted edge count is also novel: before this, it was not clear how to compare the size of node-based and edge-based circuits.
>
> > Regarding scientific novelty:
>
> Our new metrics and large-scale evaluations allow us to support multiple novel empirical claims. Previously, no single paper was able to compare all of the circuit discovery methods that we did in a systematic way (see response to comment on existing metrics). Specifically, we find that (a) edge-based circuits are better than node-based circuits (L327-328), (b) ablations from counterfactual inputs are best (L320-321), and (c) DAS (and non-basis-aligned representations more broadly) outperforms other featurizers (L439). In summary, our proposed metrics help us recover known findings (like that integrated gradients improves attribution quality), challenge others (that SAEs are not as effective as DAS at featurization), and enable new kinds of direct comparisons across methods that were previously difficult to operationalize (L432-434). We will clarify these by adding a bulleted paragraph in the introduction summarizing our contributions, and emphasizing in the paper when we obtain novel findings.
>
> The causal variable localization track includes multiple causal variables that have not been investigated in previous research. The multiple choice pointer variable was not analyzed in previous work such as [2]; while there is some evidence of a carry-the-one variable existing, e.g., [3] or [4], our baselines were unable to surpass the random performance of 50%, indicating that linear methods may not be enough to locate this variable. Finally, while the token and position variables from the IOI task were proposed in [5], we are the first to conduct experiments identifying separable sets of features for each variable. In sum, while we drew from existing datasets and tasks, our analyses have significant novelty.
>
> References:
> ===
> [1] Kusner et al. (2017). https://arxiv.org/abs/1703.06856
>
> [2] Wiegreffe et al. (2025). https://arxiv.org/abs/2407.15018
>
> [3] Kantamneni & Tegmark (2025). https://arxiv.org/pdf/2502.00873
>
> [4] Quirke et al. (2025). https://arxiv.org/pdf/2402.02619
>
> [5] Wang et al. (2023).https://arxiv.org/abs/2211.00593

---

### Decision · Program_Chairs · 2025-05-01

**Decision:**

Accept (poster)

**Comment:**

The paper develops a benchmark to evaluate MI methods. While some reviewers expressed that the use of standard datasets is a limitation of the work, all agree that this is a valuable contribution to MI, which currently lacks a standard benchmark. I also think that this is a timely contribution that can help standardize evaluations in this field and I hope the authors incorporate all comments of the reviewers in their final version.